# Identification of bipotent progenitors that give rise to myogenic and connective tissues in mouse

**Alexandre Grimaldi**[1,2,3], **Glenda Comai**[1,2], **Sebastien Mella**[4,5], **Shahragim Tajbakhsh**[1,2]*

[1]Stem Cells and Development Unit, Institut Pasteur, Paris, France; [2]UMR CNRS 3738, Institut Pasteur, Paris, France; [3]Sorbonne Universités, Complexité du Vivant, F-75005, Paris, France; [4]Cytometry and Biomarkers UTechS, Institut Pasteur, Paris, France; [5]Bioinformatics and Biostatistics Hub, Institut Pasteur, Paris, France

**Abstract** How distinct cell fates are manifested by direct lineage ancestry from bipotent progenitors, or by specification of individual cell types is a key question for understanding the emergence of tissues. The interplay between skeletal muscle progenitors and associated connective tissue cells provides a model for examining how muscle functional units are established. Most craniofacial structures originate from the vertebrate-specific neural crest cells except in the dorsal portion of the head, where they arise from cranial mesoderm. Here, using multiple lineage-tracing strategies combined with single cell RNAseq and in situ analyses, we identify bipotent progenitors expressing *Myf5* (an upstream regulator of myogenic fate) that give rise to both muscle and juxtaposed connective tissue. Following this bifurcation, muscle and connective tissue cells retain complementary signalling features and maintain spatial proximity. Disrupting myogenic identity shifts muscle progenitors to a connective tissue fate. The emergence of *Myf5*-derived connective tissue is associated with the activity of several transcription factors, including *Foxp2*. Interestingly, this unexpected bifurcation in cell fate was not observed in craniofacial regions that are colonised by neural crest cells. Therefore, we propose that an ancestral bi-fated program gives rise to muscle and connective tissue cells in skeletal muscles that are deprived of neural crest cells.

**\*For correspondence:**
shahragim.tajbakhsh@pasteur.fr

## Editor's evaluation

This study combines sophisticated lineage tracing and single-cell RNAseq analysis to provide insights into cell fate decision in myogenesis and fibrogenesis. The paper will be of interest to a broad audience of developmental biologists, as it provides evidence for a population of novel bipotent cells, which possess a signature of both muscle and connective tissue.

## Introduction

Stromal cells that are associated with skeletal muscles play critical roles in providing structural support and molecular cues (*Biferali et al., 2019*; *Kardon et al., 2003*; *Sefton and Kardon, 2019*). The majority of muscle-associated connective tissues in the head is derived from cranial neural crest cells (NCCs), an embryonic cell population that contributes to most of the structural components of the 'new head', a vertebrate innovation (*Le Douarin and Kalcheim, 1999*; *Gans and Northcutt, 1983*; *Grenier et al., 2009*; *Heude et al., 2018*; *Noden and Trainor, 2005*). Recently, the extent of this contribution was redefined in muscles derived from cranial mesoderm, including extraocular (EOM), laryngeal and pharyngeal muscles (*Comai et al., 2020*; *Grimaldi et al., 2015*; *Heude et al., 2018*;

*Kuroda et al., 2021*; *Noden and Epstein, 2010*). Interestingly, these muscles contain mesenchyme that is mesoderm-derived in their dorso-medial component, whereas the remaining muscle mass is embedded in mesenchyme that is neural crest-derived. It is unclear how the coordinated emergence of myogenic and connective tissues takes place during development, and how they establish long-lasting paracrine communication.

Along the trunk axis, paraxial somitic mesoderm gives rise to skeletal muscles and associated connective tissues (*Burke and Nowicki, 2003*). Upon signals emanating from adjacent tissues, the dermomyotome (dorsal portion of the somite) undergoes an epithelial-to-mesenchymal transition and gives rise to several cell types including all skeletal muscles of the body, vasculature, tendons and bones (*Ben-Yair and Kalcheim, 2008*; *Christ et al., 2007*). Similarly, cranial mesodermal progenitors give rise to these diverse cell types, yet, its unsegmented nature raises the question of how spatiotemporal control of these cellular identities is established. Moreover, cardiopharyngeal mesoderm, which constitutes the major portion of cranial mesoderm, has cardiovascular potential, which manifests in the embryo as regions of clonally related cardiac and craniofacial skeletal muscles (*Diogo et al., 2015*; *Swedlund and Lescroart, 2020*). This skeletal muscle/cardiac branchpoint has been the subject of intense investigation in several model organisms including ascidians, avians, and mouse (*Wang et al., 2019*). While cardiopharyngeal mesoderm was shown to give rise to connective tissues in the mammalian pharynx, the extent of its contribution to other craniofacial muscles in general has not been fully addressed (*Adachi et al., 2020*).

Recently, advanced pipelines integrating scRNAseq data and modern algorithms have been instrumental for identifying new lineage relationships during development (*Cao et al., 2019*; *He et al., 2020*; *Qiu et al., 2021*). Here, we employed lineage-restricted single-cell transcriptomics using multiple transgenic mouse lines combined with various computational methods, in situ labeling and loss-of-function experiments, and show that bipotent progenitors expressing the muscle determination gene *Myf5* give rise to both skeletal muscle and anatomically associated connective tissues. Surprisingly, this property was restricted to muscle masses lacking NCC-derived connective tissues, indicating that cranial mesoderm acts as a source of connective tissues in the absence of neural crest cells.

## Results

### Myogenic and non-myogenic mesodermal populations coexist within distinct head lineages

Somitic (*Pax3*-dependent) and cranial (*Tbx1/Pitx2-dependent*) mesoderm give rise to diverse cell types including those of the musculoskeletal system (*Figure 1A*). We first set out to explore the emergence of skeletal muscles and other associated mesodermal tissue within these programs. To that end, we employed a broad anterior mesoderm lineage-tracing strategy using the *Mesp1^{Cre/+}*;*Rosa26^{mTmG/+}* line as it labels cranial-derived mesoderm and the anterior somites (*Heude et al., 2018*). At E10.5, when craniofacial skeletal muscles start to be specified, the upper third (anterior to forelimb) of the embryos was dissected, live GFP+ cells were isolated by FACS, and processed for scRNAseq analysis (*Figure 1—figure supplement 1A-C*). After removal of doublets and lower quality cells (see Materials and methods), a large portion of the cells obtained by *Mesp1^{Cre/+}*;*Rosa26^{mTmG/+}* lineage tracing segregated as individual clusters expressing markers of adipogenic, chondrogenic, sclerotomal, endothelial, and cardiovascular lineages as well as the foregut and primitive lung mesenchyme (*Figure 1B*, *Figure 1—figure supplement 2A-B*). *Pax3*, *Pitx2*, *Tbx1*, *Myf5*, and *Myod* expression were used to identify clusters containing the cranial myogenic progenitors, annotated as 'Cardiopharyngeal mesoderm' and 'Anterior somite' (*Figure 1B–C*, *Figure 1—figure supplement 2A*).

After subsetting these clusters ('Cardiopharyngeal mesoderm' and 'Anterior somite'), a few subclusters clearly separated based on their origin and anatomical location (*Figure 1D–E*, *Figure 1—figure supplement 2C*). Surprisingly, about half of the supposedly myogenic cells exhibited a connective tissue signature, including a strong bias toward *Prrx1*, a marker of lateral plate mesoderm (*Durland et al., 2008*), *Col1a1*, a major extracellular matrix component of connective tissue cells (*De Micheli et al., 2020*), and *Twist1*, a key determinant for the mesenchymal properties of cranial mesoderm (*Bildsoe et al., 2016*; *Figure 1F*). Furthermore, the expression of *Pdgfra*, a well-defined marker of stromal cells (*Farahani and Xaymardan, 2015*), was robustly anticorrelated with the expression of

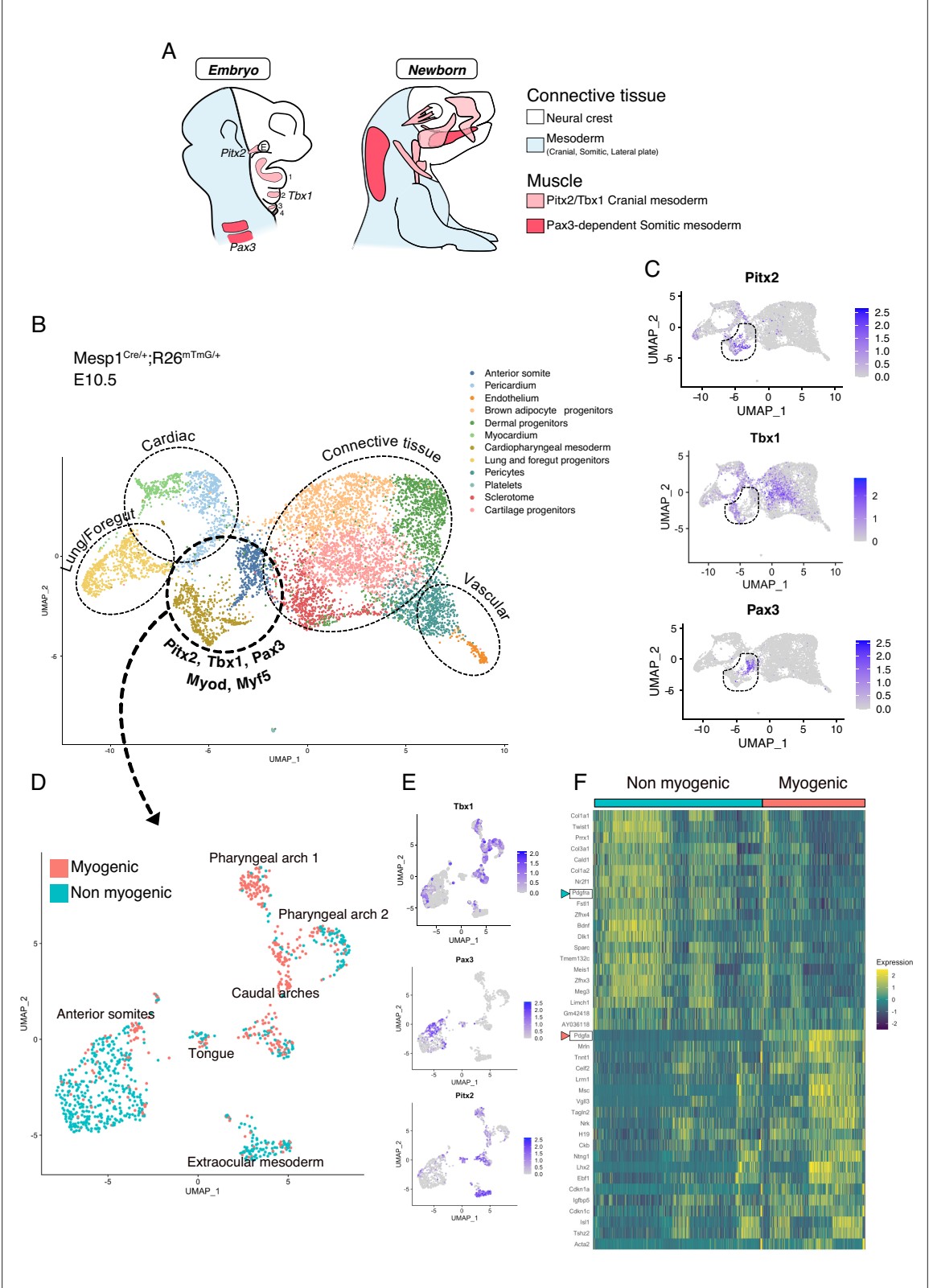

**Figure 1.** scRNAseq reveals non-myogenic populations of cranial mesoderm lineages. (**A**) Scheme of connective tissue origin in the head and known mesodermal upstream regulators. E: Eye, 1–4: Pharyngeal arches 1–4. (**B–F**) scRNAseq analysis on *Mesp1^{Cre/+}; Rosa26^{mTmG/+}* embryos at E10.5 (2 datasets of 2 embryos were aggregated to generate this data, see methods). (**B**) UMAP of *Mesp1^{Cre/+}; Rosa26^{mTmG/+}* E10.5 scRNAseq with main cell types highlighted. The clusters 'Anterior somite' and 'Cardiopharyngeal mesoderm' were subsetted for further analysis below. (**C**) UMAP expression plots of

*Figure 1 continued on next page*

*Figure 1 continued*

*Pitx2* (EOM), *Tbx1* (cranial mesoderm except EOM) and *Pax3* (somitic mesoderm), indicating the clusters of progenitors that were selected. (**D**) UMAP of progenitor subset annotated as myogenic and non-myogenic based on expression patterns found in E and F. (**E**) UMAP expression plots of *Pitx2*, *Tbx1* and *Pax3* in the *Mesp1^Cre/+^; Rosa26^mTmG/+^* E10.5 subset. (**F**) Heatmap of top 20 markers of myogenic versus non-myogenic clusters *Mesp1^Cre/+^; Rosa26^mTmG/+^* E10.5 subset. *Pdgfra/Pdgfa* genes are highlighted.

The online version of this article includes the following figure supplement(s) for figure 1:

**Figure supplement 1.** FACS strategy and preprocessing metrics of the *Mesp1*-derived E10.5 dataset.

**Figure supplement 2.** Genetic markers defining anterior mesodermal tissues.

**Figure supplement 3.** Complementary Pdgf signaling defines myogenic and non-myogenic mesodermal cells.

its ligand *Pdgfa* and associated with non-myogenic genes. Conversely, *Pdgfa,* was correlated with a myogenic cell state (*Figure 1F*, *Figure 1—figure supplement 3A-B*). Of note, myogenic *Pdgfa* expression was shown to promote adjacent sclerotomal cells to adopt a rib cartilage fate (*Tallquist et al., 2000*). Therefore, this analysis identified anatomically distinct muscle and closely associated connective tissue progenitors and highlights a potential PDGFR-mediated crosstalk between these 2 cells types.

## Transcriptional trajectories reveal a myogenic to non-myogenic cell state transition

To understand the lineage relationship between myogenic and non-myogenic cells, we exploited the unspliced and spliced variants of our scRNAseq data, and computed the RNA velocity in each cell, using a recently described tool (*Bergen et al., 2020*; *Figure 2*, *Figure 2—figure supplement 1*). RNA velocity interrogates the relative abundance of unspliced and spliced gene variants, which depends on the rates of transcription, degradation, and splicing to infer directional trajectories (*Bergen et al., 2020*; *La Manno et al., 2018*). The cell cycle status constitutes a potential bias in scRNAseq data, especially when heterogeneous populations undergo cellular expansion, commitment and differentiation (*McDavid et al., 2016*). To eliminate this potential bias, cell cycle genes were consistently regressed out during preprocessing and directional trajectories were overlaid with cell cycle phase visualization for comparisons (*Figure 2—figure supplement 1A*, Materials and methods). Notably, RNA velocity-inferred trajectories suggested that Myf5+ cells from the myogenic compartment contributed to non-myogenic cells (*Figure 2A*). These calculations were based on gene- and cluster-specific dynamics, which yield higher accuracy than the initially described RNA velocity method, while providing quantitative metrics for quality control (*Figure 2—figure supplement 1B-D* and Materials and methods).

Another powerful feature of this method is the ability to infer 'driver genes' that are responsible for most of the calculated RNA velocity, hence actively transcribed, or repressed (*Bergen et al., 2020*). Therefore, these genes can identify transitory states underlying cell fate decisions. We used this approach to uncover the driver genes that were responsible for the velocity found in anterior somites, as these cells displayed the most consistent directionality, and appeared to be independent of cell cycle (*Figure 2B*, *Figure 2—figure supplement 1A*, *Table 1*). Top transcribed driver genes included *Foxp1* (*Shao and Wei, 2018*), *Meox2* (*Noizet et al., 2016*), *Meis1* (*López-Delgado et al., 2020*), *Twist2* (*Franco et al., 2009*), *Fap* (*Puré and Blomberg, 2018*), *Pdgfra* (*Tallquist et al., 2000*), *Prrx1* (*Leavitt et al., 2020*), and *Pcolce* (*Bildsoe et al., 2016*; *Figure 2C*), which are associated with fibrosis and connective tissue development. Interestingly, we noted that *Pdgfra* appeared as a driver gene and was activated along this inferred trajectory, whereas *Pdgfa* expression decreased rapidly (*Figure 2D*). Taken together, RNA velocity analysis for anterior somite mesodermal progenitors showed that Myf5+/Pdgfa+ cells shifted toward a non-myogenic fate, which includes the downregulation of *Myf5* and *Pdgfa* and the activation of *Pdgfra* expression (*Figure 2E*).

## *Myf5*-derived lineage contributes to connective tissue cells in the absence of neural crest

Given that the number of cells examined in the EOM and pharyngeal arch mesodermal clusters from the E10.5 dataset was lower than for anterior somites, we decided to validate the relevance of Myf5-derived non-myogenic cells in these cranial regions directly in vivo. We thus examined the EOM, larynx

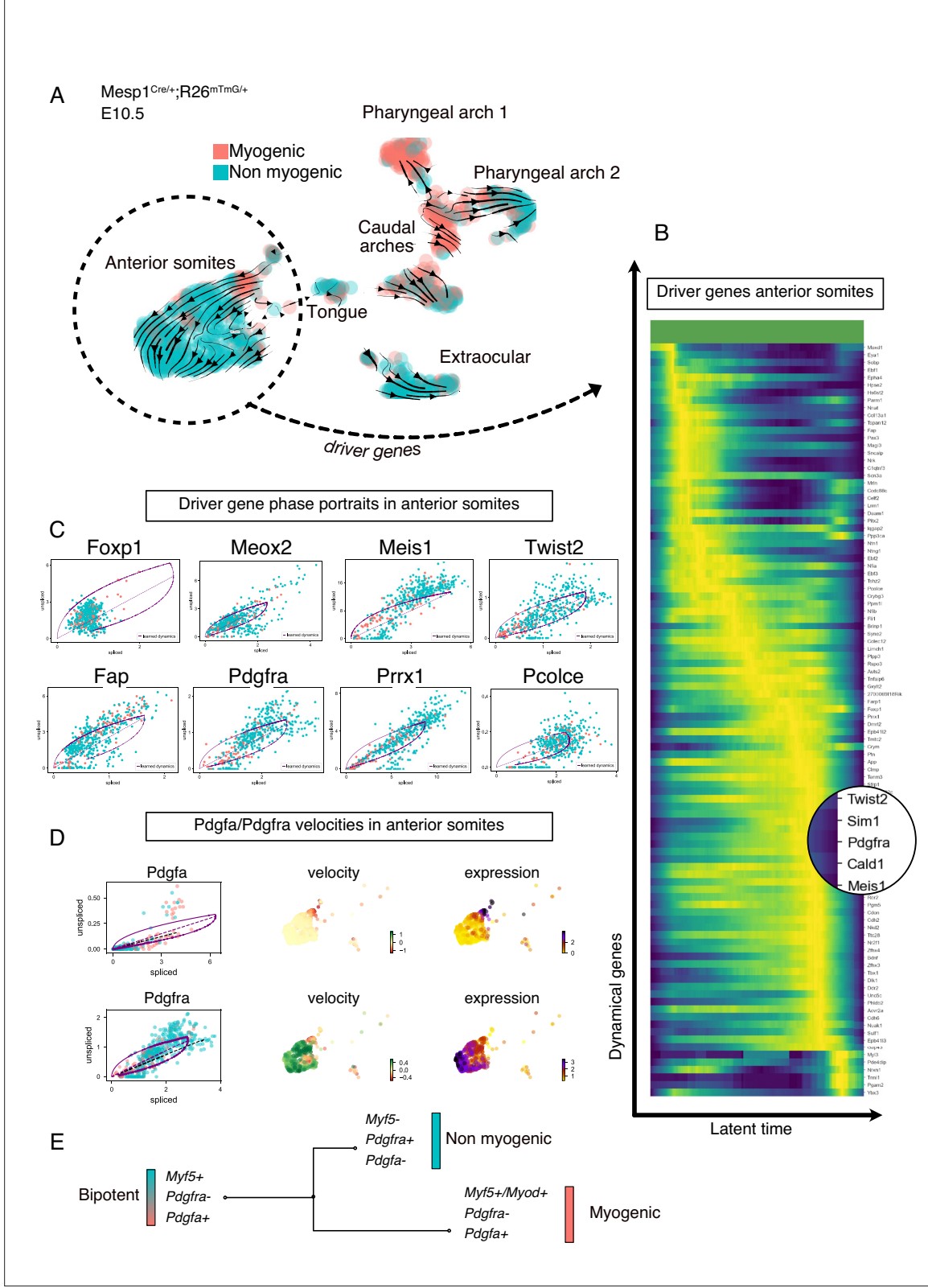

**Figure 2.** Transcriptomic dynamics reveal a myogenic to non-myogenic transition in anterior somite progenitors. (**A**) Velocity UMAP plots of *Mesp1^Cre/+^; Rosa26^mTmG/+^* embryos at E10.5 displaying myogenic and non-myogenic clusters. Arrows represent the lineage progression based on RNA velocity (relative abundance of unspliced and spliced transcripts). (**B**) Heatmap of driver genes accounting for anterior somite velocity, highlighting *Pdgfra*. Driver genes are genes that are transcriptomically active in a given cluster. (**C**) Phase portraits of few selected driver genes in the anterior somites:

*Figure 2 continued on next page*

*Figure 2 continued*

*Foxp1, Meox2, Meis1, Twist2, Fap, Pdgfra, Prrx1,* and *Pcolce.* Y-axis represents the amount of unspliced transcript per cell; X-axis represents the number of spliced transcripts per cell. A high fraction of unspliced variants indicates an active transcription of the locus, while the inverse indicates inactive/repressed transcription. Dynamics of transcription were inferred at a gene- and cluster-specific level (see Methods). (**D**) Phase portraits, RNA velocity and expression plots of *Pdgfa* and *Pdgfra* showing splicing dynamics of these two genes. (**E**) Working model of myogenic and non-myogenic fate decisions from a common bipotent progenitor in anterior somites.

The online version of this article includes the following figure supplement(s) for figure 2:

**Figure supplement 1.** Cell cycle phases and RNA velocity metrics of the *Mesp1*-derived E10.5 dataset.

and upper back muscles in the early fetus at E14.5 using a *Myf5*-lineage reporter mouse (*Myf5$^{Cre/+}$*; *Rosa26$^{TdTomato/+}$*) combined with a contemporary reporter for Pdgfra (*Pdgfra$^{H2BGFP/+}$*) (**Figure 3**). Notably, we observed GFP+ TOM+ double-positive cells in regions of EOM, laryngeal and upper back muscles that are partially or fully deprived of neural crest (**Adachi et al., 2020**; **Comai et al., 2020**; **Heude et al., 2018**; **Figure 3A–C'**). Conversely, no double-positive cells were detected in muscles that are fully embedded in neural crest derived mesenchyme such as mandibular and tongue muscles (**Heude et al., 2018**; **Figure 3D–E'**).

Mesenchymal tissues associated with the EOM arise from mesoderm in its most dorso-medial portion and from neural crest in its ventro-lateral portion (**Comai et al., 2020**; **Kuroda et al., 2021**). This dual origin makes it a prime candidate to explore the relative contribution of *Myf5*-derived cells to the associated connective tissues within a single functional unit. Using *Wnt1$^{Cre/+}$; Rosa26$^{mTmG/+}$; Myf5$^{nlacZ/+}$* (NCC tracing with *Wnt1*) and *Mesp1$^{Cre/+}$;Rosa26$^{mTmG/+}$;Myf5$^{nlacZ/+}$* (mesoderm tracing with *Mesp1*) at E13.5, we found that GFP+ cells that expressed *Myf5* (β-gal+) were exclusively present in *Mesp1*-derived domains and absent from the *Wnt1* lineage (**Figure 3F–I'**). To further evaluate the contribution of Myf5-derived cells to connective tissues in either domain, we re-examined the *Myf5$^{Cre/+}$; Rosa26$^{TdTomato/+}$; Pdgfra$^{H2BGFP/+}$*mouse line and quantified the percentage of GFP+ TOM+ cells in the EOM. As expected, we observed a medio-lateral gradient of *Myf5*-lineage contribution to EOM-associated connective tissues by E14.5, and this was anticorrelated with the local contribution of neural crest cells to connective tissues (**Figure 3J–K**). Thus, in agreement with our scRNAseq velocity analysis, these observations suggest that the mesodermal *Myf5*-lineage contributes to muscle-associated connective tissue in domains that are deprived of neural crest mesenchyme.

## Myf5-derived cells can maintain a molecular crosstalk following bifurcation into myogenic and connective tissue fates

To identify the transition between these two fates, we generated an additional sc-RNAseq dataset based on *Myf5*-lineage tracing at E11.5 (*Myf5$^{Cre/+}$;Rosa26$^{mTmG/+}$*) and produced RNA velocity streams (**Figure 4A**, **Figure 4—figure supplement 1**). We focused on the EOM and anterior somites, which were clearly distinguished as independent clusters based on the expression of *Alx4* (**Bothe and Dietrich, 2006**) and *Pax3* (**Heude et al., 2018**), respectively (**Figure 4B**). In agreement with the E10.5 mesodermal (*Mesp1*) sc-RNAseq dataset, these progenitors presented a strong dichotomy in *Pdgfa* and *Pdfgra* expression between myogenic and non-myogenic cells, respectively (**Figure 4—figure supplement 1D**). Moreover, RNA velocity suggested more myogenic to non-myogenic conversion (**Figure 4A**, **Figure 4—figure supplement 1E-H**). To explore further the cell fate transition in these regions, we used a recently described approach by creating a 'Coexpression score' based on myogenic and non-myogenic signatures (**Kameneva et al., 2021**) (see Materials and methods, **Figure 4C**). This analysis revealed that individual cells undergo a progressive switch from myogenic to non-myogenic gene expression along the inferred trajectories, where cells at the transition zone shut off the myogenic program and start activating fibrogenic genes (**Figure 4C** heatmap).

To investigate in more detail potential paracrine cell-cell communication between myogenic and non-myogenic cells, we examined their expression patterns within the EOM, given its well-defined morphology (**Comai et al., 2020**), and its strong myogenic/non-myogenic bi-directional cell-fate (**Figure 4D**). We performed single molecule fluorescent in situ hybridization (RNAscope) for *Pdgfa* and *Pdgfra* on E14.5 lineage-traced *Myf5$^{Cre/+}$;Rosa26$^{mTmG/+}$* fetuses (**Figure 4E**). In accordance with the scRNAseq analysis, we observed cells exhibiting a mostly non-overlapping, complementary pattern of *Pdgfa* and *Pdgfra* transcripts within the *Myf5*-derived lineage, while retaining anatomical proximity, even at later stages of EOM development.

**Table 1.** Driver genes underlying cell fate decisions in each dataset.

| E10.5 Anterior somites | E11.5 EOM Myogenic | E11.5 EOM Non-myogenic | E12.5 Non-myogenic | E14.5 Non-myogenic |
|---|---|---|---|---|
| Tshz2 | Ccdc141 | Zfpm2 | Mgat4c | Dnm1 |
| Eya1 | Mcm6 | Plxna4 | Cenpv | Pid1 |
| C1qtnf3 | Dync1i1 | Col23a1 | C130073E24Rik | Nrp2 |
| Meis2 | Tpm2 | Edil3 | Tbx3os1 | Ntrk3 |
| Limch1 | Celf2 | Map2 | E330013P04Rik | Tmem132c |
| Moxd1 | Sox6 | Rora | Stk26 | Egflam |
| Epha4 | Tnc | Sema5a | Edil3 | Gpr153 |
| Pitx2 | Magi3 | Colec12 | Fdft1 | Efemp1 |
| Parm1 | Sh3glb1 | Smoc1 | Lima1 | Adamts2 |
| Hpse2 | Parm1 | Ptprt | Trim59 | Brinp1 |
| Lrrn1 | Ephb1 | Ror1 | Meg3 | Vegfc |
| Dmrt2 | Bmpr1b | Dock5 | Gins3 | Twist2 |
| Myl3 | Hells | Map1b | Tpm2 | Itgb5 |
| Fap | Pdgfc | Fn1 | Cdh6 | Gria1 |
| Hs6st2 | Ptprd | Limch1 | Csmd3 | Sned1 |
| Ddr2 | Cnr1 | Tenm4 | Tceal5 | Sorcs3 |
| Cald1 | Sema3d | Rbms3 | Pclaf | Ebf2 |
| Prrx1 | Clcn5 | Srgap3 | Tspan9 | Fam19a1 |
| Magi3 | Chd7 | Tmem132c | Eps8 | Trabd2b |
| Ntn1 | Col25a1 | Sdc2 | Lmna | Plxdc2 |
| Zfhx3 | Reep1 | Add3 | Dmrt2 | Sh3gl3 |
| Meis1 | Ctnnal1 | Pdgfra | Cpeb4 | Luzp2 |
| Tnni1 | Tpm1 | Gmds | Hpgd | Pdzd2 |
| Crym | Zim1 | St6galnac3 | Rcsd1 | Sema3e |
| Ebf1 | Lmx1a | Epb41l3 | Pdgfra | Rims1 |
| Nr2f1 | Neb | Pde3a | Plac1 | Epha3 |
| Ntng1 | Atad2 | Tox | Palmd | Cyp7b1 |
| Pgm5 | Dapk2 | Smarca2 | Gucy1a1 | Gem |
| Cdh6 | Prox1 | Ctdspl | Wif1 | Ldb2 |
| Foxp1 | Lsamp | Magi2 | Naalad2 | Scube1 |
| Celf2 | Ttn | Dpysl3 | Smoc2 | Pdgfra |
| Tbx1 | Pls3 | Fgfr2 | Rassf4 | Pde1a |
| Bdnf | Slf2 | Ldb2 | Pttg1 | Nde1 |
| Colec12 | Vat1l | Igf1 | Josd2 | Enpp2 |
| Eya4 | E2f1 | Elk3 | Plxna4 | Fam107b |
| Sobp | Epb41l2 | Zmiz1 | Eya2 | Stxbp6 |
| Peg3 | Gm28653 | Dlc1 | Nrsn1 | Rerg |
| Pdgfra | Lrrn1 | Nhs | Fign | Prex2 |
| Nrk | Mef2c | Cdkn1c | Inppl1 | Man1a |
| Ptn | St8sia2 | Plpp3 | Rnf152 | Tmem45a |

*Table 1 continued on next page*

*Table 1 continued*

| E10.5 Anterior somites | E11.5 EOM Myogenic | E11.5 EOM Non-myogenic | E12.5 Non-myogenic | E14.5 Non-myogenic |
|---|---|---|---|---|
| Daam1 | Tshz1 | Ebf1 | Lasp1 | Sh3bp4 |
| Dlk1 | Wee1 | Sorbs2 | Mrln | Mcc |
| Unc5c | Slc24a3 | Baz1a | Cdt1 | Ncald |
| Lpar1 | Ncoa1 | Fat4 | Notch3 | Kdelr2 |
| Syne2 | Dek | Golgb1 | Pax3 | Pcdh19 |
| Nkd2 | Kdm5b | Hpse2 | Egfr | Gas7 |
| Brinp1 | Unc13c | Samd4 | Dbf4 | Cpt1c |
| Zfhx4 | Ddr1 | Itga9 | Bcr | Adam22 |
| Nnat | Pip4k2a | Magi1 | Mllt3 | Itgb8 |
| Gxylt2 | Fndc3c1 | Pcdh9 | Nectin1 | Dchs2 |
| Clmp | Rbm24 | Tgfbr2 | Grin3a | Cep350 |
| Ror2 | Rreb1 | Ntf3 | Cbfa2t3 | Oat |
| Nfia | Rragd | Col11a1 | Cdh2 | Rab30 |
| Ebf2 | Acsl3 | Runx1t1 | Anln | Aff2 |
| Ednra | Acvr2a | Tnrc18 | Ccdc6 | Gna14 |
| Fli1 | Zeb1 | Crym | Mcu | Slc29a1 |
| Tspan12 | Rgma | Fap | Fnip2 | Pls3 |
| Ttc28 | Arpp21 | Ppp1r1a | Kcnk13 | Traf3ip1 |
| Nfib | Lef1 | Tes | Sned1 | Rcsd1 |
| Ccdc88c | Nr2f2 | Bicc1 | Nde1 | Lgr4 |
| Col13a1 | Foxo1 | Il1rapl1 | Hipk3 | Zfp9 |
| 2700069I18Rik | Pdzrn4 | Alcam | Arhgap11a | Hs3st5 |
| Pcolce | Hmga2 | 2700069I18Rik | Fam8a1 | Aspn |
| Scn3a | Lurap1l | Dab2 | Kif21a | Nrxn1 |
| Acvr2a | Pkig | Cntln | Mtss1 | Rrm1 |
| Auts2 | Ncl | Clmn | Abcd2 | Igfbp7 |
| Col3a1 | CT025619.1 | Rbms1 | Irx5 | Slc35f3 |
| Gap43 | Erbb4 | Tmem2 | Pacs2 | Kif15 |
| Mrln | Cdk14 | Cdh6 | Nab1 | Slc1a3 |
| Pax3 | Kif21a | Lypd6 | Ccnd2 | Bmp6 |
| Sim1 | Zfp704 | Mmp2 | Bok | Dkk2 |
| Epb41l2 | Nasp | Kif5c | Dok5 | Tspan9 |
| Ppp3ca | Plekha5 | Cadm2 | Ncapg | Ets1 |
| Tnfaip6 | Cap2 | Prkg2 | Rfx8 | Gria3 |
| Tmem132c | Snca | Cped1 | Fhod3 | Sox8 |
| Tmem2 | Epha4 | Dtl | Tk1 | Melk |
| Epb41l3 | Atad5 | Ror2 | Asf1b | Ntm |
| Crybg3 | Cntn3 | Utrn | Tek | Synpo2l |
| Nrxn1 | Cacna2d1 | Foxp1 | Arfgef3 | Hlf |
| Farp1 | Pak3 | L3mbtl3 | Rnf182 | Adamts5 |

*Table 1 continued on next page*

Table 1 continued

| E10.5 Anterior somites | E11.5 EOM Myogenic | E11.5 EOM Non-myogenic | E12.5 Non-myogenic | E14.5 Non-myogenic |
|---|---|---|---|---|
| Sulf1 | Megf10 | Cdh23 | Kif14 | Plcb4 |
| Tmtc2 | Tnnt1 | Negr1 | 1810041L15Rik | Cdc25b |
| Pde4dip | Acta2 | Hmcn1 | Rrm2 | Mgat4a |
| Phldb2 | Barx2 | Col26a1 | Fgf5 | Mdfic |
| Plpp3 | Mrln | Fbn2 | Barx2 | Trpc5 |
| Ybx3 | Pgm5 | Ankrd12 | Fli1 | Kif4 |
| Ppm1l | Fmr1 | Lhfp | Jph2 | Plce1 |
| Twist2 | Smc4 | Hs3st3b1 | Dtx4 | Il17rd |
| Nuak1 | Clmp | Adgrl3 | Ncald | Mmp16 |
| Tgfb2 | Alpk2 | Svil | Zic4 | Hhip |
| Sfrp1 | Kctd1 | Mob3b | Dlc1 | Tpx2 |
| Sncaip | Meg3 | Trabd2b | Cdc45 | Ndc80 |
| Tenm3 | Samd5 | Rmst | Gatm | Bub1b |
| Cdh2 | Nrk | Prrx1 | Ssc5d | Hmmr |
| Iqgap2 | Piezo2 | 5330434G04Rik | Phactr2 | Kank4 |
| App | Robo1 | Zfhx3 | Ppp1r14c | Tmeff2 |
| Pgam2 | Col1a2 | Foxp2 | Agl | Nr4a1 |
| Rspo3 | Cntrl | Mpp6 | Tox3 | Aurkb |
| Cdon | Mllt3 | Crispld1 | Aurka | Lrrtm3 |
| Ebf3 | Peg3 | Eya1 | Cdh15 | Cenpq |

Gene set enrichment analysis of EOM myogenic and non-myogenic driver genes showed that transmembrane receptor protein kinase and SMAD activity were shared terms between the two clusters, suggesting that specific complementary signaling networks could be actively maintained between these two populations (*Figure 4—figure supplement 2A*). Both signaling pathways were reported to act as inhibitors of myogenic differentiation and could therefore be associated with progenitor cell maintenance (*Arnold et al., 2020*; *Cossu et al., 2000*). Notably, *Bmpr1b* and *Ephb1* were among the top 100 driver genes of the myogenic EOM compartment, suggesting that myogenic commitment is associated with upregulation of these kinase receptors in the EOM (*Figure 4—figure supplement 2B*, *Table 1*). Strikingly, two of their respective ligands, *Bmp4* and *Efnb1*, were preferentially expressed in non-myogenic cells. Analysis of their expression patterns in E14.5 embryos by RNAscope validated these complementary expression patterns in adjacent muscle and connective tissue domains (*Figure 4F–H*). These results favor a model where paracrine signaling networks operate between myogenic and non-myogenic *Myf5*-derived cells (*Figure 4I*), while their cellular juxtaposition is maintained through fetal stages.

## Obstructing myogenesis expands connective tissue formation from bipotent cells

The directional trajectories identified by RNA velocity in the EOM at E11.5 showed a strong bidirectionality in fate with a higher velocity confidence index at each end of the myogenic and non-myogenic domains, and lower at their interface (*Figure 5—figure supplement 1A*). This suggested that cells at the interface are bipotential while cells located on either side of this central region are committed either to a myogenic-or non-myogenic fate. To identify the regulatory factors underlying this potential bipotency, we used SCENIC, a regulatory network inference algorithm (*Aibar et al., 2017*). This tool allows regrouping of sets of correlated genes into regulons (each regulon consists of a transcription factor and its targets) based on binding motifs and co-expression. The top regulons of

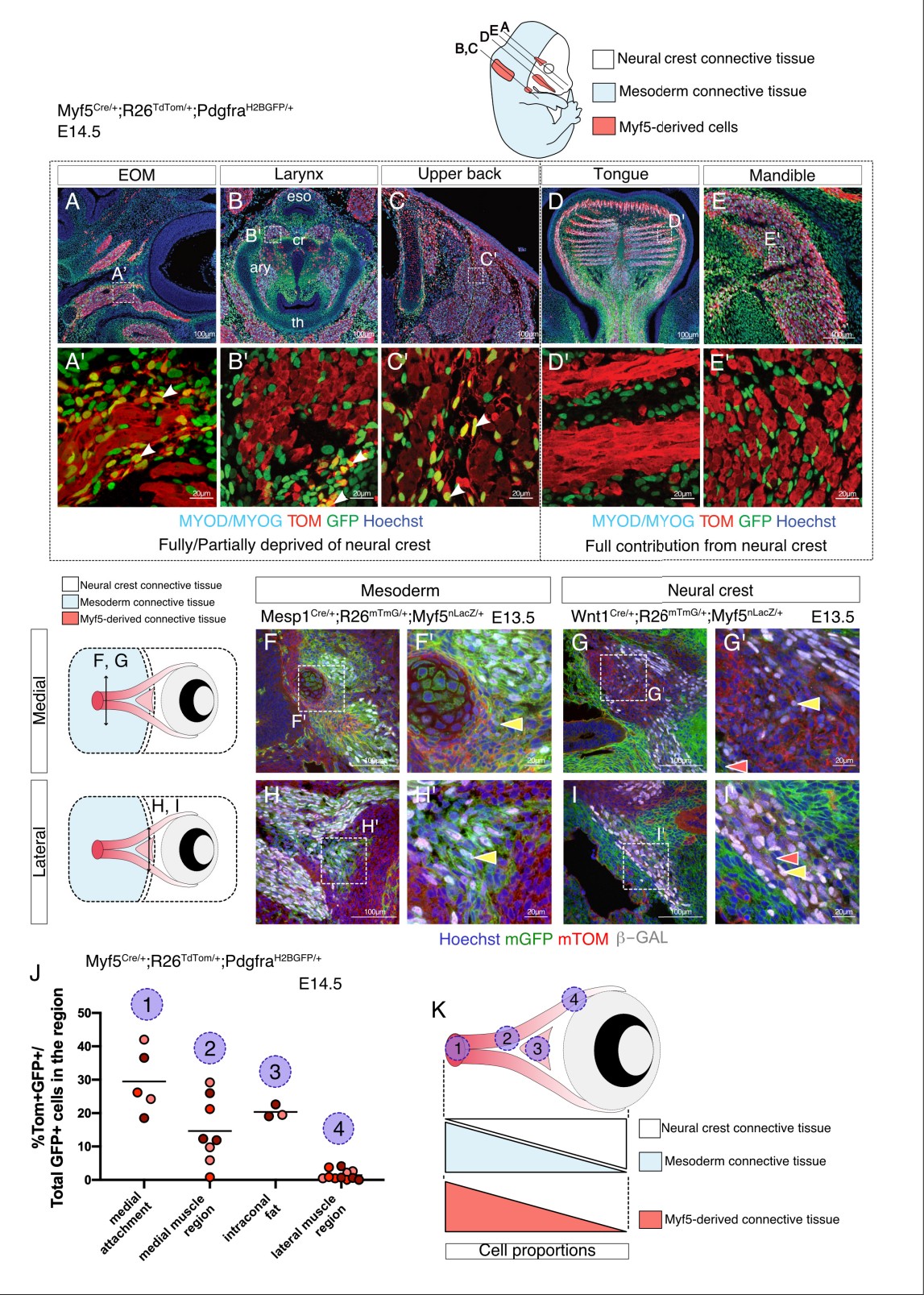

**Figure 3.** Myf5-derived mesodermal connective tissue partially compensates for the lack of neural crest. (**A-E'**) Transverse sections of an E14.5 *Myf5^{Cre/+}*; *Rosa26^{TdTomato/+}; Pdgfra^{H2BGFP/+}* embryo immunostained for Myod/Myog. White arrowheads indicate cells double-positive GFP/TOM and negative for Myod/Myog (n = 3 embryos). (**F-I'**) Transverse cryosections of the EOM at E13.5 of *Wnt1^{Cre/+}; Rosa26^{mTmG/+}; Myf5^{nlacZ/+}* (**G,I**) and *Mesp1^{Cre/+}; Rosa26^{mTmG/+}; Myf5^{nlacZ/+}* (**F,H**) immunostained for β-gal, at the level of the medial attachment (**F,G**) and lateral muscle masses (**H,I**). Yellow arrowheads indicate

*Figure 3 continued on next page*

*Figure 3 continued*

Myf5-expressing cells in the context of mesodermal and neural crest lineages. Note that Myf5-expressing cells are mGFP⁺ in the Mesp1 lineage and mGFP⁻ in the Wnt1 lineage. Red arrowheads indicate neural-crest cells that are excluded from the *Myf5* lineage (n = 2 embryos for each). (**J**) Scatter plots of the proportion of double positive cells in E14.5 *Myf5^Cre/+^; Rosa26^TdTomato/+^; Pdgfra^H2BGFP/+^* embryos in various regions throughout the EOM (the line is the mean, each dot is a tissue section, each color is a different embryo, n = 3 embryos). (**K**) Scheme highlighting the quantified regions in (**J**) and summarising the contribution of each population to periocular connective tissues. TOM: TdTOMATO.

The online version of this article includes the following source data for figure 3:

**Source data 1.** Excel table summarizing the quantification displayed on *Figure 3J*.

this analysis revealed active transcription factors underlying myogenic and non-myogenic cell fates in the EOM at E11.5. Notably, *Myf5*, *Pitx1*, *Mef2a*, and *Six1*, transcription factors known to be implicated in myogenic development (**Buckingham and Rigby, 2014**), appeared among the top regulons in myogenic cells whereas *Fli1*, *Ebf1*, *Ets1*, *Foxc1*, *Meis1*, and *Six2*, genes known for their involvement in adipogenic, vascular, mesenchymal and tendon development (**Jimenez et al., 2007**; **López-Delgado et al., 2020**; **Noizet et al., 2016**; **Truong and Ben-David, 2000**; **Whitesell et al., 2019**; **Yamamoto-Shiraishi and Kuroiwa, 2013**), constituted some of the highly active non-myogenic transcription factors (**Figure 5A**). Interestingly, recent work uncovered *Fli1* as a potential regulator of vascular fate in multipotent myogenic progenitors (**Ferdous et al., 2021**). Accordingly, we found that *Scube1*, a gene known for its involvement in vasculature development, was upregulated in the Pdgfra+ non-myogenic fraction of the EOM (**Figure 5—figure supplement 1B**). RNAscope in situ hybridization confirmed these findings and showed that *Scube1* was expressed at the level of the EOM medial attachment at E14.5 (**Figure 5—figure supplement 1C-E**). In addition, *Scube1* was reported to promote BMP signaling (**Liao et al., 2016**). Thus, the EOM tendon attachment seems to rely on transcription factors and markers that are typically vascular, hence suggesting that some of them are coopted.

Given that *Myf5* appeared itself as a top myogenic regulon (**Figure 5A**), we interrogated the fate of *Myf5*-expressing progenitors in a *Myf5^nlacZ/nlacZ^* null embryos at E12.5 (**Figure 5B–E'**). As previously reported, the EOMs are absent in this mutant (**Figure 5C**, asterisk) (**Sambasivan et al., 2009**). Interestingly, some β-gal+ cells were found in the cartilage primordium (Sox9+) of the EOM in the heterozygous control indicating that cells with recent *Myf5* activity diverged to this non-myogenic fate (**Figure 5D–D'**). Notably, disruption of *Myf5* activity led to a threefold increase in the proportion of non-myogenic *Myf5*-derived cells in this region (**Figure 5E–F**). In contrast, no double-positive cells were found in the masseter, a muscle fully embedded in neural crest-derived connective tissue, even in the absence of *Myf5* (**Figure 5F**). *Myf5* expression is thus necessary to maintain a balance between myogenic and non-myogenic cell fates of Myf5+ progenitors only in neural crest-depleted regions. In contrast, very few Pdgfra+ cells were found to be derived from *Myod* expressing cells in *Myod^iCre^;Rosa26^TdTomato/+^;Pdgfra^H2BGFP/+^* fetuses at E14.5, particularly in the EOM and the back muscles (about 3 and 1.5 cells per 100 μm² of muscle, respectively)(**Figure 5G, I, J**). Accordingly, the masseter lacked *Myod*-derived connective tissue cells (**Figure 5H and J**). These observations indicate that progenitors that bifurcate to myogenic and non-myogenic cell fates are present only in neural-crest depleted regions. This property is associated primarily with *Myf5* expression, as subsequent activation of *Myod* within this lineage appears to lock cell fate into the myogenic program and suppress their connective tissue potential (**Figure 5K**).

## Myf5-derived connective tissues are observed in fetal stages

Although we identified *Myf5*-derived non-myogenic cells in various regions of the embryo, it was not clear if this population was continuously generated throughout development. To address this issue, we performed two more scRNAseq experiments at E12.5 and E14.5, using contemporary *Myf5* labeling, which led to much fewer non-myogenic cells that could be captured (*Myf5^GFP-P/+^*; **Figure 6**, **Figure 6—figure supplement 1**, **Figure 6—figure supplement 2**). In accordance with the earlier datasets, some cells that appeared to belong to muscle anlagen of EOM, somites and caudal arches progressed toward a non-myogenic state (**Figure 6A–C'**). To assess the identity of these cells, we performed a gene set enrichment network analysis combining the differentially expressed genes of non-myogenic clusters of all stages. We found that all stages contributed relatively equally to each 'GO Molecular Function' and 'Reactome pathways' terms despite their relatively diverse gene expression signatures (**Figure 6D–E'**, **Figure 6—figure supplement 3**). This suggests that these non-myogenic cells are

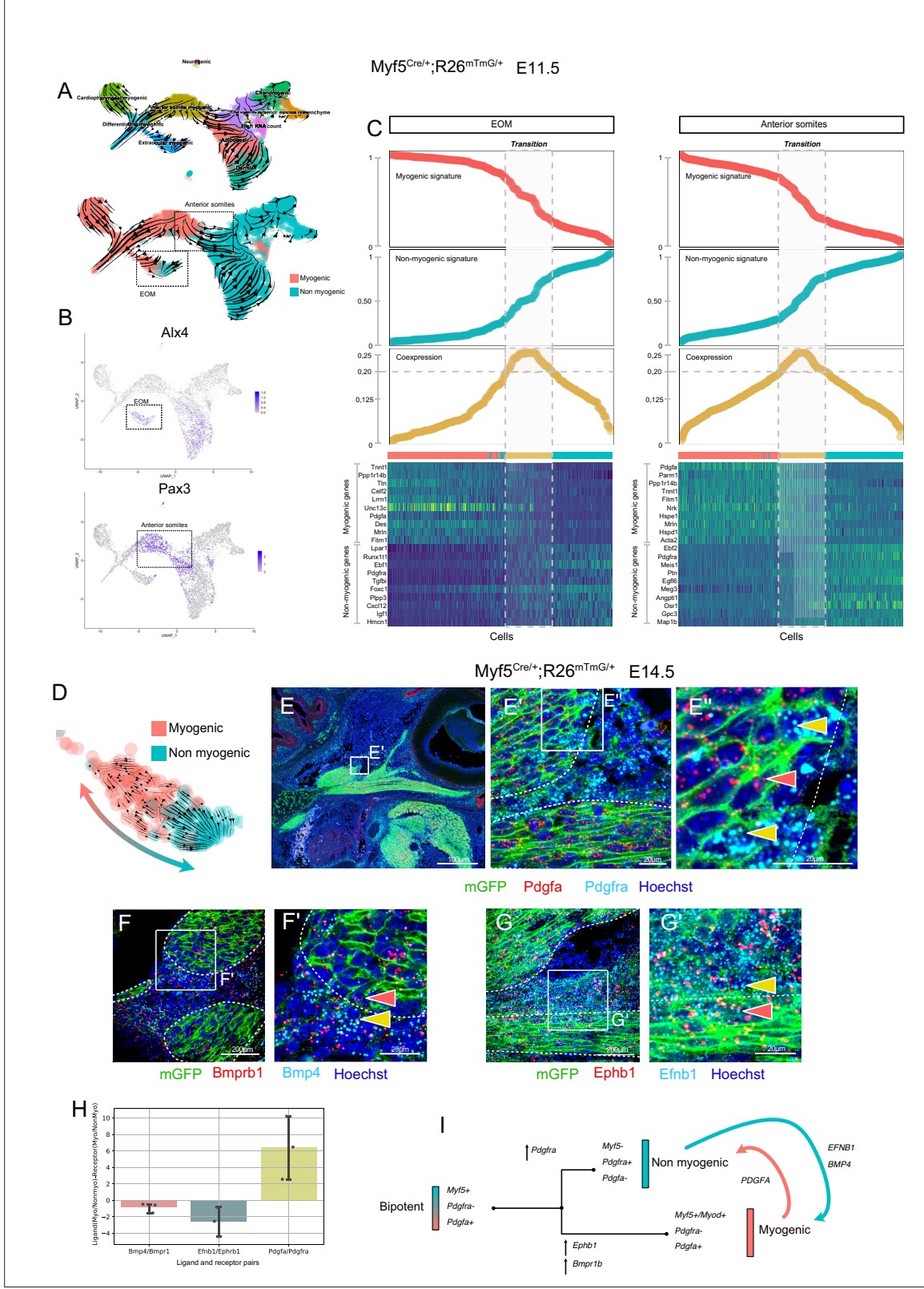

**Figure 4.** Maintenance of signaling cues between Myf5-derived myogenic and non-myogenic cells in EOM. (**A–D**) scRNAseq analysis of the *Myf5*<sup>Cre/+</sup>; *Rosa26*<sup>mTmG/+</sup> E11.5 dataset (2 datasets of 2 embryos were aggregated to generate this data, see Materials and methods). (**A**) UMAPs of *Myf5*<sup>Cre/+</sup>; *Rosa26*<sup>mTmG/+</sup> E11.5 RNA velocity trajectories. (**B**) Expression plots of *Alx4* and *Pax3*, highlighting EOM and Anterior somite clusters, respectively. (**C**) Plots of Myogenic and Non-myogenic signatures, Coexpression score and heatmaps of top markers, highlighting the transition population in EOM and

*Figure 4 continued on next page*

*Figure 4 continued*

anterior somites. Cells are ordered based on their non-myogenic signature score (increasing). The coexpression score is the product of the myogenic and non-myogenic signatures. Cells presenting a coexpression score higher than 0.20 are highlighted in yellow. These cells represent the transition between the myogenic and non-myogenic fates. (**D**) UMAP of the EOM subset revealing the bipartite fate of *Myf5*-expressing cells. (**E-G'**) RNAscope on *Myf5^Cre/+^; Rosa26^mTmG/+^* E14.5 tissue sections with *Pdgfra* (cyan) and *Pdgfa* (red) probes (**E-E''**), *Bmprb1* (red) and *Bmp4* (cyan) probes (**F-F'**) and *Ephb1* (red) and *Efnb1* (cyan) probes (**G-G'**). *Myf5*-derived cells are labelled by membrane GFP staining (n = 3 embryos each). Red and yellow arrowheads indicate *Myf5*-derived myogenic and non-myogenic cells respectively. The dotted lines highlight the boundary of the muscle masses. (**H**) Quantification of the Ligand-Receptor scores for each pair (see Materials and methods). Note that these ratios are negative in the case of Bmp and Eph (signaling from non-myogenic to myogenic) but positive for Pdgf (signaling from myogenic to non-myogenic). (**G**) Model of myogenic and non-myogenic cell communication following bifurcation from a bipotent cell.

The online version of this article includes the following source data and figure supplement(s) for figure 4:

**Source data 1.** Excel table summarizing the quantification displayed on *Figure 4H*.

**Figure supplement 1.** FACS strategy, preprocessing metrics, expression patterns, and RNA velocity metrics of the *Myf5*-derived E11.5 dataset.

**Figure supplement 2.** Kinase signaling complementarity in the EOM at E11.5.

relatively homogeneous in gene signatures throughout cranial muscles. Highly significant terms hinted at a myogenic-supporting role, providing muscle progenitors with extracellular matrix components, and contributing to neuronal guidance (*Figure 6E*). Among these terms, presence of Pdgf signalling and receptor kinase activity indicated, once again, that the interactions found in the EOM could occur also at later stages in various craniofacial muscles that are deprived of neural crest derived connective tissue.

## A novel regulatory network underlies the non-myogenic cell fate

Myf5+ bipotent progenitors were observed at multiple stages and anatomical locations, and they yielded a relatively homogeneous population expressing common markers associated with extracellular matrix components, cell adhesion molecules, and kinase signalling. To assess whether the regulatory mechanisms guiding this transition are distinct in different locations in the head, we set out to explore the common molecular switches underlying cell fate decisions. To do so, we developed a pipeline where we combined the list of driver genes at the start of the non-myogenic trajectory (*Table 1*) with the most active regulons in the non-myogenic region (Materials and methods, code in open access). This resulted in a network consisting of the most active transcription factors and the most transcriptionally dynamic genes found at the non-myogenic branchpoint. We performed this operation for each dataset independently and displayed them as individual networks (*Figure 7—figure supplement 1A-D*). Finally, we overlapped the list of these 'driver regulators' to identify the common transcription factors guiding the non-myogenic cell fate decision (*Figure 7A*). Notably, *Foxp2*, *Hmga2*, *Meis1*, *Meox2*, and *Tcf7l2* were identified in all four scRNAseq datasets as key driver regulators, and thus are likely to play significant role in the non-myogenic transition (*Figure 7A*, *Table 2*).

Forkhead box transcription factors FOXC1 and FOXC2 were reported to regulate the balance between myogenic and vascular lineages within somites (*Lagha et al., 2009*; *Mayeuf-Louchart et al., 2016*). Interestingly, *Foxc1* has been reported to promote both cranial vasculature and cranial cartilage development in zebrafish (*Whitesell et al., 2019*; *Xu et al., 2021*). FOXP2 immunostaining on *Myf5^Cre/+^;Rosa26^TdTom/+^;Pdgfra^H2BGFP/+^* E12.5 embryos showed that the *Myf5*-derived EOM cells expressed a relatively high level of *Foxp2* compared to mandible and trunk muscles, consistent with their apparent high contribution to connective tissue (*Figure 7B–E*).

To gain further insights into the transitioning population, we performed FACS analysis of dissected head, limb and trunk regions of *Myf5^Cre/+^;Rosa26^TdTom/+^;Pdgfra^H2BGFP/+^* embryos at E12.5 (*Figure 7F–G*). We focused on TOM+ cells (Myf5-lineage) and assessed their GFP expression levels as a readout of their commitment toward connective tissue. This analysis identified non-FAPs cells (GFP^low^) transitioning towards a Pdgfra+ state in head and trunk regions but very few in the limb (*Figure 7G*). Interestingly, while trunk muscles presented the largest portion of transitioning cells (40%), a similar transitioning population was noted in the head (20%) despite a large contribution of NCC to head connective tissues. Thus, cardiopharyngeal mesoderm may have a superior potential to give rise to connective tissue compared to somite-derived progenitors in the limb (1.5%).

In addition, Tcfs and Lef1 were among the top common regulators identified, and they form a complex effector for the canonical Wnt pathway. Previous work showed that during cranial myogenesis,

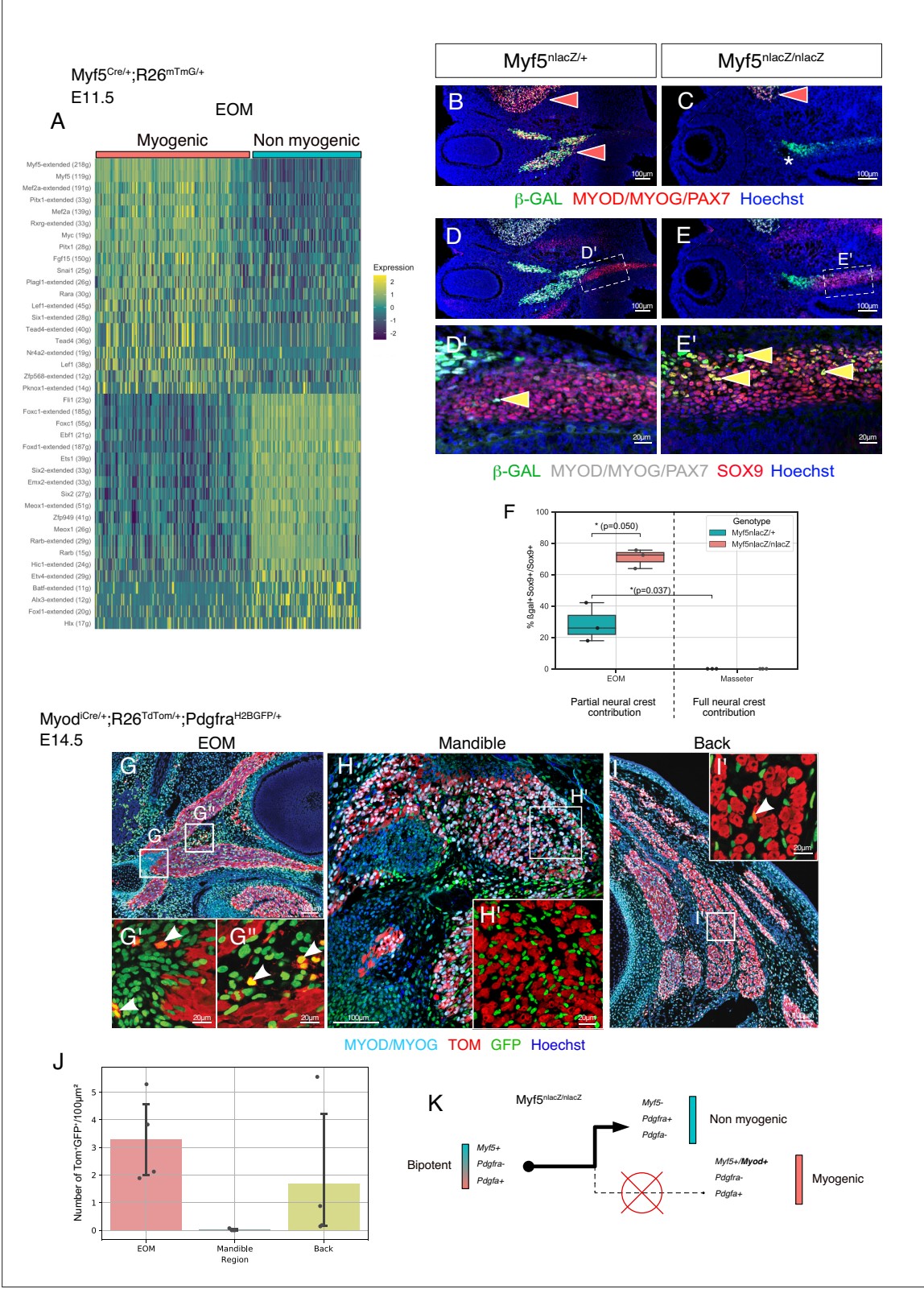

**Figure 5.** Disruption of *Myf5* increases the connective tissue output from bipotent cells. (**A**) Heatmap of top regulons (transcription factor and associated targets) of the EOM subset of the *Myf5^Cre/+^; Rosa26^mTmG/+^* E11.5 dataset. The suffix '_extended' indicates that the regulon includes motifs that have been linked to the TF by lower confidence annotations, for instance, inferred by motif similarity. Number in brackets indicates number of genes comprising the regulon (n = 2 pooled datasets). (**B–C**) Transverse sections of *Myf5^nlacZ/+^* (**B**), and *Myf5^nlacZ/nlacZ^* (**C**) embryos in the EOM region at E12.5

*Figure 5 continued on next page*

*Figure 5 continued*

immunostained for β-gal (green), and Myod/Myog/Pax7 (red). Red arrowheads indicate β-gal/ Myod/Myog/Pax7 double positive cells in control EOM/ Masseter and in mutant Masseter. Asterisk highlights the lack of myogenic progenitors in the EOM region of the mutant embryo, indicated by the absence of Myod/Myog/Pax7 staining. (D-E') Transverse sections of *Myf5^{nlacZ/+}* (D-D'), and *Myf5^{nlacZ/nlacZ}* (E-E') in the EOM region at E12.5 immunostained for β-gal (green), Sox9 (red), and Myod/Myog/Pax7 (gray). Yellow arrowheads indicate β-gal/Sox9 double positive cells and show an expansion of this cell population in the mutant. (F) Quantification of proportion of β-gal+;Sox9+ double positive cells in the total Sox9+ population of the EOM and Masseter muscles. Each dot is a different sample, the center line of the boxplot is the median value. (n = 3 embryos, p-values were calculated using a two-sided Mann-Whitney U test). (G-I') Transverse sections of *Myod^{iCre/+}; Rosa26^{TdTomato/+}; Pdgfra^{H2BGFP/+}* embryos at E14.5 immunostained for Myod/ Myog (committed and differentiating myoblasts) in the extraocular (G-G''), mandibular (H-H'), and back muscles (I-I'). White arrowhead indicates double positive cells (GFP+ TOM+). (J) Quantification of double positive cells (GFP+ TOM+) in EOM, mandibular muscles and back muscles per 100 μm² area on *Myod^{iCre/+}; Rosa26^{TdTomato/+}; Pdgfra^{H2BGFP/+}* sections shown in E-G (n = 4 embryos). (K) Model of lineage progression from bipotent cells in a *Myf5* null background.

The online version of this article includes the following source data and figure supplement(s) for figure 5:

**Source data 1.** Excel table summarizing the quantification displayed on *Figure 5F*.

**Source data 2.** Excel table summarizing the quantification displayed on *Figure 5J*.

**Figure supplement 1.** The vascular marker *Scube1* is expressed in *Myf5*-derived non-myogenic cells in the EOM.

**Figure supplement 1—source data 1.** Excel table summarizing the quantification displayed on *Figure 5—figure supplement 1E*.

neural crest cells release inhibitors of the Wnt pathway to promote myogenesis (*Tzahor et al., 2003*). It is thus tempting to speculate that in the absence of neural crest, mesoderm-derived progenitors can give rise to connective tissue by maintaining canonical Wnt activity. To test this hypothesis, we examined the expression of Axin2, a common readout for Wnt/β-cat activity (*Babb et al., 2017*; *van de Moosdijk et al., 2020*). Interestingly, Axin2 levels were elevated in the non-myogenic portion of all the different datasets (*Figure 7—figure supplement 1E-H*). Additionally, Dkk2, which has been described as an activator of Wnt/β-cat pathway in the neural crest (*Devotta et al., 2018*), was also found to be elevated, indicative of a putative positive-feedback loop mechanism supporting the maintenance of this population.

## Discussion

Distinct fates can emerge through the specification of individual cell types, or through direct lineage ancestry from bipotent or multipotent cells. Here, we addressed this issue in the context of the emergence of myogenic and associated connective tissue cells during the formation of craniofacial muscles. By combining state-the-art computational methods and in-situ analyses, we identified the transcriptional dynamics, the intercellular communication networks, and the regulators controlling the balance between complementary cell fates. Specifically, our work provides evidence for a novel mesoderm-derived bipotent cell population that gives rise to muscle and associated connective tissue cells spatiotemporally, and only in regions deprived of neural crest cells (*Figure 7—figure supplement 2*).

Brown adipocytes, neurons, pericytes, and rib cartilage have been reported to express *Myf5* in ancestral cells (*Daubas et al., 2000*; *Haldar et al., 2008*; *Sebo et al., 2018*; *Stuelsatz et al., 2014*). Interestingly, when *Myf5* expression is disrupted, cells can acquire non-myogenic fates and contribute to connective tissue (this study), cartilage, and dermis (*Tajbakhsh et al., 1996*), while others remain apparently undifferentiated (cells labeled with an asterisk in *Figure 5C*). It is likely that these cells are undergoing apoptosis as reported previously (*Sambasivan et al., 2009*). These studies suggest that *Myf5*-expression alone is not sufficient to promote robust myogenic fate in multiple regions of developing embryos. Consistent with these observations, Myod+ cells do not contribute to rib cartilage (*Wood et al., 2020*) and give rise to few connective tissue cells in the periocular and back regions (this study). These findings are also consistent with the role of *Myod* in defining the committed myogenic cell state and its higher chromatin-remodelling capacity compared to Myf5 (*Conerly et al., 2016*; *Tapscott, 2005*). In contrast to a previous study (*Stuelsatz et al., 2014*), we found no neural-crest derived cells expressing *Myf5* during EOM tissue genesis at E13.5 (using *Wnt1^{Cre/+};Rosa26^{mTmG/+};Myf5^{nlacZ/+}*). We note that *Myf5*-expressing cells contribute to non-myogenic cells from early embryonic stages (E10.5) and continue to do so in the fetus, indicating that these bipotent cells persist well after muscles are established.

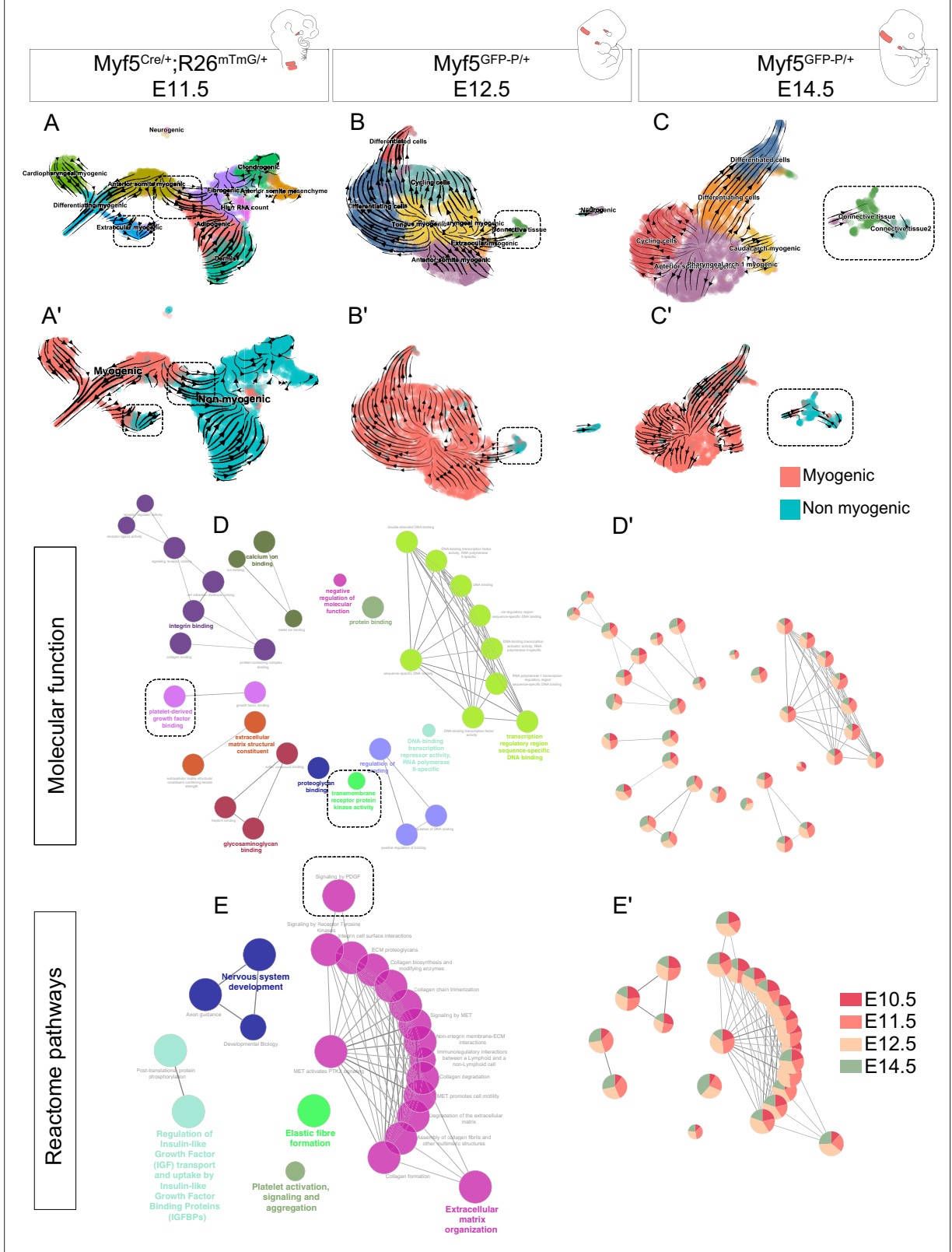

**Figure 6.** Myf5-derived non-myogenic cells are generated continuously up to fetal stages. (**A-C'**) RNA velocity plots of *Myf5^Cre/+^; Rosa26^mTmG/+^* E11.5, *Myf5^GFP-P/+^* E12.5 and *Myf5^GFP-P/+^* E14.5 datasets (n = 2 pooled datasets, n = 1 embryo and n = 1 embryo, respectively) displaying cell-type annotation (**A–C**) and myogenic and non-myogenic clustering (**A'-C'**). The dotted boxes highlight the transitions to non-myogenic clusters in each dataset. (**D–E**) Gene ontology network of GO Molecular Function and Reactome pathway performed on combined top 100 markers using Cluego. These terms

*Figure 6 continued on next page*

*Figure 6 continued*

were generated using the sum of all differentially expressed genes of the non-myogenic clusters across all datasets (see Materials and methods). (**D'**-**E'**) Relative contribution of each stage to term node represented as piecharts (i.e. the proportion of genes underlying this term coming from that stage). Dotted boxes highlight the shared tyrosine kinase and PDGF signaling pathways.

The online version of this article includes the following figure supplement(s) for figure 6:

**Figure supplement 1.** FACS strategy, preprocessing metrics, expression patterns, and RNA velocity metrics in the *Myf5*^GFP-P/+^E12.5 dataset.

**Figure supplement 2.** FACS strategy, preprocessing metrics, expression patterns and RNA velocity metrics in the *Myf5*^GFP-P/+^ E14.5 dataset.

**Figure supplement 3.** Non-myogenic *Myf5*-derived cells display a similar gene ontology.

Here, we also identifed a core set of transcription factors specifically active in the non-myogenic cells across all datasets. We propose that these genes guide bipotent cells to a non-myogenic fate and thus confer mesenchymal properties to non-committed progenitors. Recent studies have identified anatomically distinct fibroblastic populations using single-cell transcriptomics, yet unique markers could not be identified (*Muhl et al., 2020*; *Sacchetti et al., 2016*), making characterisation of cell subtypes challenging. Tcf4/Tcf7l2 was identified as a master regulator of fibroblastic fate during muscle-associated connective tissue development, although it is also expressed in myogenic progenitors at lower levels (*Kardon et al., 2003*; *Mathew et al., 2011*; *Sefton and Kardon, 2019*). We also report that this gene is one of the main regulators of connective tissue fate. Other transcription factors have been linked to skin fibroblast fates including *Tcf4*, *Six2*, *Meox2*, *Egr2*, and *Foxs1*, and their repression favors a myofibroblastic potential (*Noizet et al., 2016*). *Six2* and *Meox2* were also identified in our analysis, which raises the question of the shared genetic programs between myofibroblastic cells and fibroblastic cells derived from progenitors primed for myogenesis during development.

Interestingly, *Prrx1*, a marker for lateral plate mesoderm (*Durland et al., 2008*), was differentially expressed in the connective tissue population at various stages. Although lateral plate mesoderm is identifiable in the trunk, its anterior boundaries in the head are unclear (*Prummel et al., 2020*). More detailed analyses of *Prrx1*, *Isl1,* and *Myf5* lineages need to be carried out to delineate the specific boundaries of each progenitor contribution to cranial connective tissues.

Kinase receptors have been implicated in a number of developmental programs for both muscle and associated connective tissues (*Arnold et al., 2020*; *Knight and Kothary, 2011*; *Olson and Soriano, 2009*; *Tallquist et al., 2000*; *Tzahor et al., 2003*; *Vinagre et al., 2010*). For example, the differentiation of fetal myoblasts is inhibited by growth factors Tgfβ and Bmp4 (*Cossu et al., 2000*). Epha7 signaling is active in embryonic and adult myocytes and promotes their differentiation (*Arnold et al., 2020*). Significantly, we noticed a striking and lasting complementary expression of *Pdgfa* and *Pdgfra* throughout embryonic stages, in the myogenic and non-myogenic progenitors respectively. Pdgf ligands emanating from hypaxial myogenic cells under the control of *Myf5* were shown to be necessary from rib cartilage development (*Tallquist et al., 2000*; *Vinagre et al., 2010*). Additionally, Pdgfra promotes expansion of fibroblasts during fibrosis (*Olson and Soriano, 2009*). Interestingly, we found that *Pdgfa* expression was reduced in cells expressing high levels of *Myog* at the fetal stage (*Figure 6—figure supplement 2C*). Therefore, *Myf5*-derived myogenic progenitor cells might guide non-myogenic *Myf5*-derived expansion, which in turn provides ligands and extracellular matrix components to favor myogenic development and patterning. Moreover, unlike trunk myogenesis, cranial muscle development relies on the expression of Wnt and Bmp inhibitors from surrounding tissues (*Tzahor et al., 2003*). Interestingly, we showed that the *Myf5*-derived non-myogenic cells express *Bmp4*, *Dkk2*, and *Axin2*. Additionally, we showed that the Wnt effector complex *Tcf/Lef* is expressed to a lower extent in these cells. It is thus likely that these cells maintain their non-myogenic fate by promoting Bmp production and Wnt activity cell-autonomously.

Of note, another study suggested shared fate relationships between fibroblast connective tissue cells and skeletal muscle where fibroblastic cells commit to myogenic fate during limb development (*Esteves de Lima et al., 2021*). Regarding the possibility that some non-myogenic cells may retain bipotent characteristics, our data suggests that the opposite is true during cranial muscle development. First, RNA velocity analysis did not reveal transitioning cells from non-myogenic clusters to myogenic (even at early stages), nor do they express myogenic markers. Further, at least some of these non-myogenic cells gave rise to chondrocytes, which to our knowledge has never been shown to give rise to skeletal muscle. Additionally, bipotency appears to be more associated with myogenic

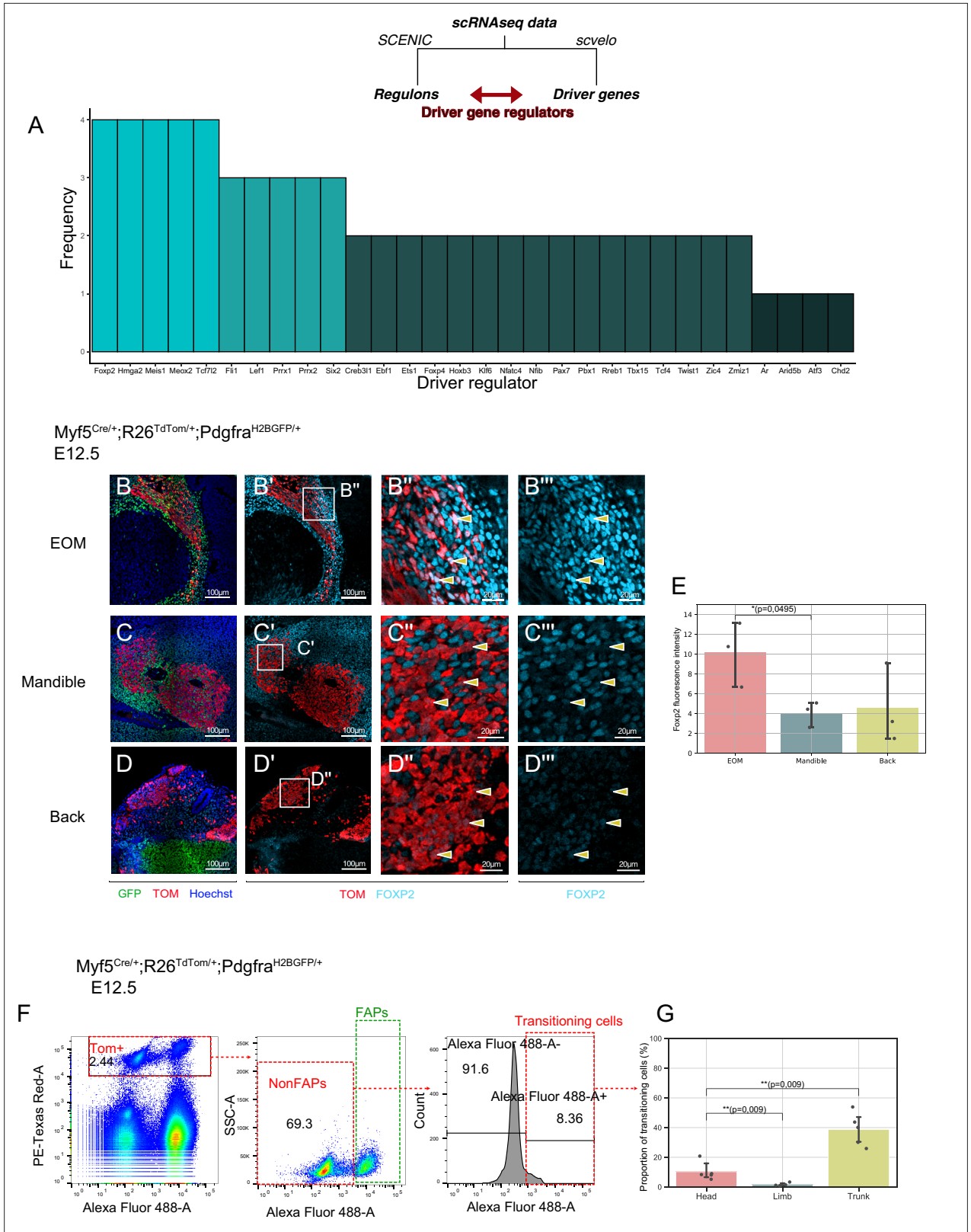

**Figure 7.** A shared program involving Forkhead-box transcription factors supports non-myogenic fate transition at various stages and anatomical locations. (**A**) Barplot displaying frequency of appearance of most predominant transcription factors as driver regulators (4 = present in all four datasets as driver regulon, 1 = present in a single dataset). (**B-D"**) Transverse sections of an E12.5 *Myf5^{Cre/+}*; *Rosa26^{TdTomato/+}*; *Pdgfra^{H2BGFP/+}* embryo immunostained for Foxp2 at the level of the EOM (**B-B"**), Mandibular muscles (**C-C"**), and Back muscles (**D-D"**). Yellow arrowheads indicated the double positive cells

*Figure 7 continued on next page*

*Figure 7 continued*

to better appreciate Foxp2 intensity in *Myf5*-derived cells. (**E**) Quantification of Foxp2 signal intensity in TOM+ (*Myf5*-derived) cells in each muscle (n = 3 embryos). Statistical test performed: Mann-Whitney U test. (**F**) FACS plots of dissected E12.5 *Myf5*^Cre/+^; *Rosa26*^TdTomato/+^; *Pdgfra*^H2BGFP/+^ embryos (head region here) highlighting the *Myf5*-derived GFP- TOM+ population transitioning to the GFP+ TOM+ population. Each plot was generated on the population gated in the previous one ('Singlets', 'TOM+' and 'NonFaps'). FAPS:Fibroadipogenic progenitors, a denomination for resident Pdgfra+ cells. (**G**) Quantification of the transitioning population in Head, Limb and Trunk. Proportion of transitioning cells is calculated as the number of Alexa488+/ Total cell number in the 'NonFAPs' gate. Note that the Head region is mostly populated by muscles embedded in neural crest (n = 5 embryos). TOM: TdTOMATO.

The online version of this article includes the following source data and figure supplement(s) for figure 7:

**Source data 1.** Excel table summarizing the quantification displayed on *Figure 7E*.

**Source data 2.** Excel table summarizing the quantification displayed on *Figure 7G*.

**Figure supplement 1.** Wnt/β-cat positive feedback loop may promote non-myogenic cell fate.

**Figure supplement 2.** Model of Myf5+ bipotent progenitors giving rise to muscle and associated connective tissues.

cells since they express *Myf5*, and to a minor extent *Myod*. Finally, we did not observe NCC-derived *Myf5*+ cells indicating that connective tissue in the head does not give rise to muscle. Nevertheless, to formally exclude the possibility of connective tissue progenitors giving rise to muscle in the embryo, analysis of appropriate markers would need to be done (ex. *Pdgfra*-driven lineage). Further studies should provide insights into the evolutionary ancestry of progenitors that bifurcate to give rise to myogenic and connective tissue cells by studying other model organisms that are devoid of neural crest cells.

# Materials and methods

## Key resources table

| Reagent type (species) or resource | Designation | Source or reference | Identifiers | Additional information |
|---|---|---|---|---|
| Strain, strain background (*Mus musculus*) | B6D2F1/JRj | Janvier | | |
| Genetic reagent (*M. musculus*) | *Myf5*^Cre^ | PMID:17418413 | MGI:3710099 | Dr. Mario R Capecchi (Institute of Human Genetics, University of Utah, USA) |
| Genetic reagent (*M. musculus*) | *Wnt1*^Cre^ | PMID:9843687 | MGI:J:69326 | Pr. Andrew P. McMahon (Keck School of Medicine of the University of Southern California, USA) |
| Genetic reagent (*M. musculus*) | *Mesp1*^Cre^ | PMID:10393122 | MGI:2176467 | Pr. Yumiko Saga (National Institute of Genetics, Japan) |
| Genetic reagent (*M. musculus*) | *Myf5*^nlacZ^ | PMID:8918877 | MGI:1857973 | Dr. Shahragim Tajbakhsh (Department of Developmental and Stem Cell Biology, Institut Pasteur, France) |
| Genetic reagent (*M. musculus*) | *Rosa26*^tdTomato^ | PMID:20023653 | MGI:3809524 | Dr. Hongkui Zeng (Allen Institute for Brain Science, USA) |
| Genetic reagent (*M. musculus*) | *Rosa26*^mT/mG^ | PMID:17868096 | MGI:3716464 | Pr. Philippe Soriano (Icahn School of Medicine at Mt. Sinai, USA) |
| Genetic reagent (*M. musculus*) | *Pdgfra*^H2BGFP^ | PMID:12748302 | MGI:2663656 | Pr. Philippe Soriano (Icahn School of Medicine at Mt. Sinai, USA) |
| Genetic reagent (*M. musculus*) | *Myod*^iCre^ | PMID:19464281 | MGI:3840216 | Pr. David Goldhamer (University of Connecticut, USA) |
| Genetic reagent (*M. musculus*) | *Myf5*^GFP-P^ | PMID:15386014 | MGI:3055340 | Dr. Shahragim Tajbakhsh (Department of Developmental and Stem Cell Biology, Institut Pasteur, France) |
| Chemical compound, drug | Sucrose,for molecular biology, ≥ 99.5% (GC) | Sigma-Aldrich | S0389-500G | |
| Chemical compound, drug | Gelatin | Sigma-Aldrich | G-7041 | |

*Continued on next page*

*Continued*

| Reagent type (species) or resource | Designation | Source or reference | Identifiers | Additional information |
|---|---|---|---|---|
| Antibody | Anti-Foxp2 5C11A8 (Mouse monoclonal) | Santa Cruz | SC-517261 | IF (1:200) |
| Antibody | Anti-β-gal (Chicken polyclonal) | Abcam | Cat. #: ab9361 RRID:AB_307210 | IF (1:1000) |
| Antibody | Anti-β-gal (Rabbit polyclonal) | MP Biomedicals | Cat. #: MP 559761 RRID:AB_2687418 | IF (1:1500) |
| Antibody | Anti-GFP (Chicken polyclonal) | Aves Labs | Cat. #: 1020 RRID:AB_10000240 | IF (1:500) |
| Antibody | Anti-GFP (Chicken polyclonal) | Abcam | Cat. #: 13970 RRID:AB_300798 | IF (1:1000) |
| Antibody | Anti-Myod (Mouse monoclonal) | Dako | Cat. #: M3512 RRID:AB_2148874 | IF (1:100) |
| Antibody | Anti-Myod (Mouse monoclonal) | BD-Biosciences | Cat. #: 554130 RRID:AB_395255 | IF (1:500) |
| Antibody | Anti-Pax7 (Mouse monoclonal) | DSHB | Cat. #: Pax7 RRID:AB_528428 | IF (1:20) |
| Antibody | Anti-Myog (Mouse monoclonal) | DSHB | Cat. #: F5D RRID:AB_2146602 | IF (1:20) |
| Antibody | Alexa Fluor 633 F(ab')2 Fragment of Goat Anti-Rabbit IgG (H + L) (polyclonal antibody) | Life Technologies | Cat. #: A-21072 RRID:AB_2535733 | IF (1:500) |
| Antibody | Alexa Fluor 555 F(ab')2 Fragment of Goat Anti-Rabbit IgG (H + L) (polyclonal antibody) | Life Technologies | Cat. #: A-21430 RRID:AB_2535851 | IF (1:500) |
| Antibody | Alexa Fluor 488 F(ab')2 Fragment of Goat Anti-Rabbit IgG (H + L) (polyclonal antibody) | Life Technologies | Cat. #: A-11070 RRID:AB_2534114 | IF (1:500) |
| Antibody | Alexa Fluor 633 Goat Anti-Chicken IgG (H + L) (polyclonal antibody) | Life Technologies | Cat. #: A-21103 RRID:AB_2535756 | IF (1:500) |
| Antibody | Alexa Fluor 488 Goat Anti-Chicken IgG (H + L) (polyclonal antibody) | Life Technologies | Cat. #: A-11039 RRID:AB_2534096 | IF (1:500) |
| Antibody | Alexa Fluor 633 Goat Anti-Mouse IgG1 (γ1) (polyclonal antibody) | Life Technologies | Cat. #: A-21126 RRID:AB_2535768 | IF (1:500) |
| Antibody | Alexa Fluor488 AffiniPure Goat Anti-Mouse IgG1 (γ1) (polyclonal antibody) | Jackson ImmunoResearch | Cat. #: 115-545-205 RRID:AB_2338854 | IF (1:500) |
| Antibody | Cy3-AffiniPure Goat Anti-Mouse IgG1 (γ1) (polyclonal antibody) | Jackson ImmunoResearch | Cat. #: 115-165-205 RRID:AB_2338694 | IF (1:500) |
| Antibody | Cy3-AffiniPure Goat Anti-Mouse IgG2a (γ2a) (polyclonal antibody) | Jackson ImmunoResearch | Cat. #: 115-165-206 RRID:AB_2338695 | IF (1:500) |
| Antibody | Dylight 405 Goat Anti-Mouse IgG2a (γ2a) (polyclonal antibody) | Jackson ImmunoResearch | Cat. #: 115-475-206 RRID:AB_2338800 | IF (1:500) |
| Commercial assay, kit | Hoechst 33,342 | Thermo Scientific | Cat. #:H3570 | |
| Commercial assay, kit | RNAscope Multiplex Fluorescent reagent Kit-V2 | ACD/Bio-techne | Cat. #: 323100 | |
| Commercial assay, kit | RNAscope H202 & Protease Plus Reagents | ACD/Bio-techne | Cat #: 322330 | |
| Commercial assay, kit | Opal 650 Reagent Pack | PerkinElmer | Cat. #: FP1496001KT | 1:1,500 of reconstituted reagent in RNAscope Multiplex TSA Buffer |
| Commercial assay, kit | Opal 570 Reagent Pack | PerkinElmer | Cat. #: FP1488001KT | 1:1,500 of reconstituted reagent in RNAscope Multiplex TSA Buffer |
| Commercial assay, kit | RNAscope Mm-Pdgfa | Advanced Cell Diagnostics, Inc | Cat #:411361 | |
| Commercial assay, kit | RNAscope Mm-Pdgfra | Advanced Cell Diagnostics, Inc | Cat #:480661-C2 | |
| Commercial assay, kit | RNAscope Mm-Bmpr1b | Advanced Cell Diagnostics, Inc | Cat #:533941 | |

*Continued*

| Reagent type (species) or resource | Designation | Source or reference | Identifiers | Additional information |
|---|---|---|---|---|
| Commercial assay, kit | RNAscope Mm-Efnb1 | Advanced Cell Diagnostics, Inc | Cat #:526761 | |
| Commercial assay, kit | RNAscope Mm-Bmp4-O1-C3 | Advanced Cell Diagnostics, Inc | Cat #:527501-C3 | |
| Commercial assay, kit | RNAscope Mm-Ephb1-C3 | Advanced Cell Diagnostics, Inc | Cat #:567571-C3 | |
| Commercial assay, kit | RNAscope Mm-Scube1 | Advanced Cell Diagnostics, Inc | Cat #:488131 | |
| Chemical compound, drug | Paraformaldehyde | Electron Microscopy Sciences | Cat. #: 15710 | |
| Chemical compound, drug | Isopentane | VWR | Cat. #: 24872.298 | |
| Chemical compound, drug | Triton X-100 | Sigma | Cat. #: T8787 | |
| Chemical compound, drug | Tween 20 | Sigma | Cat. #: P1379 | |
| Chemical compound, drug | TrypLE | ThermoFisher | Cat #: 12604013 | |
| Chemical compound, drug | Calcein Blue | eBioscience | Cat #: 65-0855-39 | |
| Chemical compound, drug | Propidium Iodide | ThermoFisher | Cat #: P1304MP | |
| Commercial assay, kit | Chromium Next GEM Chip G Single Cell Kit, 16 rxns | 10 X Genomics | Cat #: 1000127 | |
| Commercial assay, kit | Chromium Next GEM Single Cell 3' GEM, Library & Gel Bead Kit v3.1, 4 rxns | 10 X Genomics | Cat #:1000128 | |
| Commercial assay, kit | NextSeq 500/550 High Output Kit v2.5 | Illumina | Cat #: 20024906 | |
| Commercial assay, kit | Agilent High Sensitivity DNA Kit | Agilent | Cat #:5067–4626 | |
| Commercial assay, kit | Agilent High Sensitivity DNA Reagents | Agilent | Cat #:5067–4627 | |
| Commercial assay, kit | Qubit dsDNA HS Assay Kit | Life Technologies | Cat #:Q32854 | |
| Software, algorithm | RStudio | Rstudio | | |
| Software, algorithm | Anaconda | Anaconda Inc | | |
| Software, algorithm | Zen | Zeiss | | |
| Software, algorithm | Cytoscape | Cytoscape Team | | |
| Software, algorithm | Fiji | Johannes Schindelin, Ignacio Arganda-Carreras, Albert Cardona, Mark Longair, Benjamin Schmid, and others | | |
| Software, algorithm | Prism | GraphPad Software | | |
| Software, algorithm | FlowJo | FlowJo | | |

## scRNAseq data generation

For E10.5 to E12.5 embryos, the cranial region above the forelimb was dissected in ice-cold 3% FBS in PBS and mechanically dissociated with forceps and pipetting. The same procedure was applied at E14.5 but the dissection was refined to the pharyngeal and laryngeal regions. Tissues were then digested in TrypLE (ThermoFisher Cat #: 12604013) during 3 rounds of 5 min incubation (37 °C, 1400 RPM), interspersed with gentle pipetting to further dissociate the tissue. Cells were resuspended in FBS 3%, filtered, and incubated with Calcein Blue (eBioscience, Cat #: 65-0855-39) and Propidium Iodide (ThermoFisher Cat #: P1304MP) to check for viability. Viable cells were sorted on BD FACS Aria III and manually counted using a hemocytometer. RNA integrity was assessed with Agilent Bioanalyzer 2,100 to validate the isolation protocol prior to scRNAseq (RIN >8 was considered acceptable). A total of 4000–13,000 cells were loaded onto 10 X Genomics Chromium microfluidic chip and cDNA libraries were generated following manufacturer's protocol. Concentrations and fragment sizes were measured using Agilent Bioanalyzer and Invitrogen Qubit. cDNA libraries were sequenced using NextSeq 500 and High Output v2.5 (75 cycles) kits. Genome mapping and count matrix generation were done following 10X Genomics Cell Ranger pipeline.

**Table 2.** Driver regulators of non-myogenic fate in each dataset.

| | E10.5 | E11.5 | E12.5 | E14.5 |
|---|---|---|---|---|
| Foxp2 | (+) | (+) | (+) | (+) |
| Hmga2 | (+) | (+) | (+) | (+) |
| Meis1 | (+) | (+) | (+) | (+) |
| Meox2 | (+) | (+) | (+) | (+) |
| Tcf7l2 | (+) | (+) | (+) | (+) |
| Fli1 | (+) | (+) | (+) | (-) |
| Lef1 | (-) | (+) | (+) | (+) |
| Prrx1 | (+) | (+) | (-) | (+) |
| Prrx2 | (-) | (+) | (+) | (+) |
| Six2 | (+) | (+) | (+) | (-) |
| Creb3l1 | (-) | (+) | (-) | (+) |
| Ebf1 | (+) | (-) | (+) | (-) |
| Ets1 | (-) | (+) | (-) | (+) |
| Foxp4 | (+) | (+) | (-) | (-) |
| Hoxb3 | (-) | (+) | (+) | (-) |
| Klf6 | (-) | (+) | (-) | (+) |
| Nfatc4 | (-) | (+) | (-) | (+) |
| Nfib | (-) | (+) | (+) | (-) |
| Pax7 | (-) | (-) | (+) | (+) |
| Pbx1 | (-) | (+) | (-) | (+) |
| Rreb1 | (-) | (-) | (+) | (+) |
| Tbx15 | (+) | (+) | (-) | (-) |
| Tcf4 | (+) | (-) | (+) | (-) |
| Twist1 | (+) | (+) | (-) | (-) |
| Zic4 | (+) | (-) | (+) | (-) |
| Zmiz1 | (-) | (+) | (+) | (-) |
| Ar | (-) | (-) | (-) | (+) |
| Arid5b | (-) | (-) | (+) | (-) |
| Atf3 | (-) | (-) | (-) | (+) |
| Chd2 | (+) | (-) | (-) | (-) |

(+): Present, (-): Absent.

## RNA velocity and driver genes

RNA velocity analyses were performed using scvelo (*Bergen et al., 2020*) in Python. This tool allows inferring velocity flow and driver genes using scRNAseq data, with major improvements from previous methods (*La Manno et al., 2018*). First, unspliced and spliced transcript matrices were generated using velocyto (*La Manno et al., 2018*) command line function, which outputs unspliced, spliced, and ambiguous matrices as a single loom file. These files were combined with filtered Seurat objects to yield objects with unspliced and spliced matrices, as well as Seurat-generated annotations and cell-embeddings (UMAP, tSNE, PCA). These datasets were then processed following scvelo online guide and documentation. Velocity was calculated based on the dynamical model (using *scv.tl.recover_dynamics( adata)*, and *scv.tl.velocity(adata, mode='dynamical')*) and when outliers were detected, differential kinetics based on top driver genes were calculated and added to the model (using *scv. tl.velocity(adata, diff_kinetics = True)*). Specific driver genes were identified by determining the top likelihood genes in the selected cluster. The lists of top 100 drivers for each stage are given in *Table 1*.

## Data processing

scRNAseq datasets were preprocessed using Seurat in R (https://satijalab.org/seurat/) (*Butler et al., 2018*). Cells with more than 20% of mitochondrial gene fraction were discarded. The number of genes expressed averaged to 4000 in all four datasets. Dimension reduction and UMAP generation were performed following Seurat workflow. Doublets were inferred using DoubletFinder v3 (*McGinnis et al., 2019*). Cell cycle genes, mitochondrial fraction, number of genes, number of UMI were regressed in all datasets following Seurat dedicated vignette. We noticed that cell cycle regression, although clarifying anatomical diversity, seemed to induce low and high UMI clustering (*Figure 4A*, *Figure 4—figure supplement 1C*). For the E10.5 and E11.5 datasets, two replicates were generated from littermates and merged after confirming their similitude. For subsequent datasets (E12.5 and E14.5), no replicates were used. Annotation and subsetting were also performed in Seurat. 'Myogenic' and 'Non-myogenic' annotations were based on *Pdgfa* and *Pdgfra* expression and myogenic genes *Myf5*, *Myod*, and *Myog*. Cells not expressing *Pdgfa* were annotated as 'non-myogenic' unless they express myogenic genes. Cells expressing *Pdgfa* were annotated as 'myogenic'. We noticed that at later stages, *Pdgfa* expression decreases in Myog+ cells. Driver genes of connective tissue at E12.5 and E14.5 were determined using cluster annotations obtained from Leiden-based clustering. Myogenic and non-myogenic scores were generated by aggregating the total expression of all genes

in a signature based on the top 10 markers of these compartments (visible on *Figure 4C*). Each score was then divided by the sum of the two to generate myogenic and non-myogenic signatures. The coexpression score was defined by the product of these signatures. To generate the plots, cells were ordered based on their non-myogenic signature. The 'transition' was defined as cells with a coexpression score higher than 0.20.

## Gene regulatory network inference

Gene regulatory networks were inferred using SCENIC (R implementation) (*Aibar et al., 2017*) and pySCENIC (Python implementation) (*Van de Sande et al., 2020*). This algorithm allows regrouping of sets of correlated genes into regulons (i.e. a transcription factor and its targets) based on motif binding and co-expression. UMAP and heatmap were generated using regulon AUC matrix (Area Under Curve) which refers to the activity level of each regulon in each cell. We used two cisTarget databases: 'mm9-500bp-upstream-7species.mc9nr' (500 bp upstream of TSS) and 'mm9-tss-centered-10kb-7species.mc9nr' (10kb ±TSS).

## Driver regulons

Results from SCENIC and scvelo were combined to identify regulons that could be responsible for the transcriptomic induction of driver genes. Similarly to the steps mentioned above, SCENIC lists of regulons were used to infer connections between transcription factors and driver gene. Networks were generated as explained above and annotated with 'Active regulon' or 'driver gene'. The lists of individual driver regulons of each dataset were then combined and the most recurring driver regulons were identified. The code is available at this address: https://github.com/TajbakhshLab/DriverRegulators, (copy archived at swh:1:rev:49db57e7ede9f248de937b7a47eb96b02aa2ce67; *Grimaldi, 2021*).

## Gene set enrichment analysis

Gene set enrichment analyses were performed on either the top markers (obtained from Seurat function FindAllMarkers) or from driver genes (obtained from scvelo), using Cluego (*Bindea et al., 2009*). 'GO Molecular Pathway', 'GO Biological Process' and 'Reactome pathways' were used independently to identify common and unique pathways involved in each dataset. In all analyses, an enrichment/depletion two-sided hypergeometric test was performed and p-values were corrected using the Bonferroni step down method.

## Mouse strains

Animals were handled as per European Community guidelines and the ethics committee of the Institut Pasteur (CETEA) approved protocols (APAFIS#6354–20160809 l2028839). The following strains were previously described: $Myf5^{Cre}$ (*Haldar et al., 2008*), $Myod^{iCre}$ (*Kanisicak et al., 2009*), $Mesp1^{Cre}$ (*Saga et al., 1999*), $Tg:Wnt1Cre$ (*Danielian et al., 1998*), $Rosa26^{TdTom}$ (Ai9; *Madisen et al., 2010*), $Rosa26^{mTmG}$ (*Muzumdar et al., 2007*), $Myf5^{nlacZ}$ (*Tajbakhsh et al., 1996*), $Pdgfra^{H2BGFP}$ (*Hamilton et al., 2003*) and $Myf5^{GFP-P}$ (*Kassar-Duchossoy et al., 2004*). To generate $Myf5^{Cre/+};Rosa26^{TdTomato/+};Pdgfra^{H2BGFP/+}$ embryos, $Myf5^{Cre/+}$ females were crossed with $Pdgfra^{H2BGFP/+};Rosa26^{TdTomato/TdTomato}$ males. Mice were kept on a mixed genetic background C57BL/6JRj and DBA/2JRj (B6D2F1, Janvier Labs). Mouse embryos and fetuses were collected between embryonic day (E) E10.5 and E14.5, with noon on the day of the vaginal plug considered as E0.5.

## Immunofluorescence

Collected embryonic and adult tissues were fixed 2.5 h in 4% paraformaldehyde (Electron Microscopy Sciences, Cat #:15710) in PBS with 0.2–0.5% Triton X-100 (according to their stage) at 4 °C and washed overnight at 4 °C in PBS. In preparation for cryosectioning, embryos were equilibrated in 30% sucrose in PBS overnight at 4 °C and embedded in OCT. Cryosections (16–20 μm) were left to dry at RT for 30 min and washed in PBS. For Foxp2 immunostaining (Santa Cruz Cat. #: SC-517261), embryos were first equilibrated in 15% sucrose overnight, then in a 15% sucrose/7.5% gelatin solution at 37 °C the next day and embedded in the same solution the following day. Blocks were then kept at 4 °C in a humid environment and trimmed, before being submerged in liquid nitrogen-cooled isopentane at –60 °C to freeze. After cryosectioning, slides were washed twice for 15 min each at 37 °C inPBS to remove the gelatin. The primary antibodies used in this study are chicken polyclonal

anti-β-gal (Abcam, Cat #: ab9361, dilution 1:1000), mouse monoclonal IgG1 anti-Myod (BD Biosciences, Cat# 554130, dilution 1:100), mouse monoclonal IgG1 anti-Pax7 (DSHB, Cat. #: AB_528428, dilution 1:20), rabbit anti-mouse Sox9 (Millipore, Cat. #: AB5535, dilution 1/2000), rabbit polyclonal anti-Tomato (Clontech Cat. #: 632496, dilution 1:400) and chicken polyclonal anti-GFP (Abcam Cat. #: 13970, dilution 1:1000). Images were acquired using Zeiss LSM780 or LSM700 confocal microscopes and processed using ZEN software (Carl Zeiss). Control and mutant embryos were selected randomly, quantifications were performed blindly by hiding the discriminating channels. Quantifications were performed using Fiji (https://imagej.net/software/fiji/). Barplots, dotplots and boxplots were generated using Seaborn (https://seaborn.pydata.org; https://seaborn.pydata.org/) or Prism (https://www.graphpad.com/scientific-software/prism/). For the $Myf5^{Cre/+};Rosa26^{TdTomato/+};Pdgfra^{H2BGFP/+}$ embryos, 4 regions were manually defined across the medio-lateral axis. For each region, the absolute number of double-positive cells within the defined area was divided by the total number of GFP+ cells which was determined first, and blindly (with the TOMchannel disabled). For the $Myod^{iCre}$ lineage-tracing experiment, the absolute number of double positive cells were counted and divided by the area of the muscle given by the TOM channel. These 'number of cells/area' scores were then corrected based on the size of the image in microns, and adjusted to match an area of 100 μm$^2$ of muscle. To quantify the intensity of Foxp2 immunostaining, we first generated ROIs of the $Myf5$-derived cells based on the TOM channels as previously mentioned and extracted the mean pixel value. All images were acquired using the exact same settings for a given embryo.

## RNAscope in situ hybridization

Embryos for in situ hybridization were fixed overnight in 4% PFA. Embryos were equilibrated in 30% sucrose in PBS and sectioned as described for immunofluorescence. RNAscope probes Mm-Pdgfa (411361), Mm-Pdgfra (480661-C2), Mm-Bmpr1b (533941), Mm-Efnb1 (526761), Mm-Bmp4-O1-C3 (527501-C3), Mm-Ephb1-C3 (567571-C3) and Mm-Scube1 (488131) were purchased from Advanced Cell Diagnostics, Inc. In situ hybridization was performed using the RNAscope Multiplex Fluorescent Reagent Kit V2 as described previously (*Comai et al., 2019*). Quantifications were performed using Fiji (https://imagej.net/software/fiji/). 2 ROIs were first defined visually using the GFP channel: 'myogenic' and 'non-myogenic'. The channels containing the RNAscope signals were then thresholded to obtain binary images, and measurement of the 'Area%' was performed for each ROI. For each probe, we generated a ratio of myogenic to non-myogenic signal. The ratio of each receptor was then substracted from the ratio of each corresponding ligand.

## Acknowledgements

We acknowledge funding support from the Institut Pasteur, Association Française contre le Myopathies, Agence Nationale de la Recherche (Laboratoire d'Excellence Revive, Investissement d'Avenir; ANR-10-LABX-73) and MyoHead (ANR-19-CE13-0008-01), Association Française contre les Myopathies (Grant #20510), Fondation pour la Recherche Médicale (Grant # FDT201904008277), and the Centre National de la Recherche Scientifique. We gratefully acknowledge the UtechS Photonic BioImaging, C2RT, Institut Pasteur, supported by the French National Research Agency (France BioImaging; ANR-10–INSB–04; Investments for the Future).

## Additional information

### Competing interests

Shahragim Tajbakhsh: Reviewing editor, *eLife*. The other authors declare that no competing interests exist.

### Funding

| Funder | Grant reference number | Author |
|---|---|---|
| Association Française contre les Myopathies | 20510 | Alexandre Grimaldi |

| Funder | Grant reference number | Author |
|---|---|---|
| Fondation pour la Recherche Médicale | FDT201904008277 | Alexandre Grimaldi |
| Agence Nationale de la Recherche | ANR-10-LABX-73 | Shahragim Tajbakhsh |
| Centre National de la Recherche Scientifique | | Glenda Comai |
| Agence Nationale de la Recherche | ANR-19-CE13-0008-01 | Shahragim Tajbakhsh |

The funders had no role in study design, data collection and interpretation, or the decision to submit the work for publication.

### Author contributions

Alexandre Grimaldi, Conceptualization, Data curation, Formal analysis, Funding acquisition, Investigation, Methodology, Project administration, Software, Supervision, Validation, Visualization, Writing – original draft, Writing – review and editing; Glenda Comai, Conceptualization, Formal analysis, Investigation, Validation, Visualization, Writing – review and editing; Sebastien Mella, Data curation, Formal analysis, Software, Writing – review and editing; Shahragim Tajbakhsh, Conceptualization, Funding acquisition, Project administration, Resources, Supervision, Writing – review and editing

### Author ORCIDs

Alexandre Grimaldi ⓘ http://orcid.org/0000-0002-5978-2057
Glenda Comai ⓘ http://orcid.org/0000-0003-3244-3378
Sebastien Mella ⓘ http://orcid.org/0000-0002-8679-5718
Shahragim Tajbakhsh ⓘ http://orcid.org/0000-0003-1809-7202

### Ethics

Animals were handled as per European Community guidelines and the ethics committee of the Institut Pasteur (CETEA) approved protocols (APAFIS#6354-20160809 l2028839).

### Decision letter and Author response

Decision letter https://doi.org/10.7554/eLife.70235.sa1
Author response https://doi.org/10.7554/eLife.70235.sa2

## Additional files

### Supplementary files

• Transparent reporting form

### Data availability

scRNAseq datasets are available in open access on DRYAD at the following address: https://data-dryad.org/stash/dataset/doi:10.5061/dryad.gf1vhhmrs?. The code that was used to generate the driver regulators is available at this address: https://github.com/TajbakhshLab/DriverRegulators, (copy archived at swh:1:rev:49db57e7ede9f248de937b7a47eb96b02aa2ce67). Source data files have been provided for Figure 3J, Figure 4H, Figure 5F, Figure 5J, Figure 5-figure supplement 1E, Figure 7E and Figure 7G.

The following dataset was generated:

| Author(s) | Year | Dataset title | Dataset URL | Database and Identifier |
|---|---|---|---|---|
| Grimaldi A, Mella S | 2022 | scRNAseq_raw_filtered_preprocessed | https://dx.doi.org/10.5061/dryad.gf1vhhmrs | Dryad Digital Repository, 10.5061/dryad.gf1vhhmrs |

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
