## [Editor Report]

This study combines sophisticated lineage tracing and single-cell RNAseq analysis to provide insights into cell fate decision in myogenesis and fibrogenesis. The paper will be of interest to a broad audience of developmental biologists, as it provides evidence for a population of novel bipotent cells, which possess a signature of both muscle and connective tissue.

---

## [Decision Letter]

**Decision letter after peer review:**

Thank you for submitting your article "Identification of bipotent progenitors that give rise to myogenic and connective tissues in mouse" for consideration by *eLife*. Your article has been reviewed by 3 peer reviewers, and the evaluation has been overseen by a Reviewing Editor and Marianne Bronner as the Senior Editor. The following individual involved in review of your submission has agreed to reveal their identity: Peter Currie (Reviewer #2).

Essential revisions:

The reviewers found the paper interesting and potentially acceptable after revisions, including better descriptions of the scRNAseq by refining some of the in silico analyses and providing better quantitation. Currently, there is too much emphasis placed on possible signaling cross-talk which are not validated. While functional validation would be helpful, this seems like a minor aspect of the paper which could be reduced and toned down. The main contribution is the identification of myf5 positive progenitors able to contribute to the fibroblastic lineage. The full comments of the reviewers are provided below to help you in revising the paper.

*Reviewer #1 (Recommendations for the authors):*

Overall, this is a very interesting study and the conclusions of the paper are consistent with their presented data. The work will be of interest for the readership of *eLife*.

There are some concerns I have that require some additional clarification:

Overall, the description of the scRNAseq and the velocity analysis could be improved and for instance the criteria to decide whether a population is myo or fibrogenic should be better explained in the text.

In figure 1B, lung and foregut precursors are identified whereas the population was sorted based on Mesp1 previous expression which labels the mesoderm. Are these mesodermal components of the lung and foregut?

The myogenic nature of the cells could be better described and illustrated on figure 1 and 2 and the expression pattern of myf5 in the different clusters would help. It would also be helpful to show the expression pattern of PDGFRA on the UMAPs as this is a key gene in this study. The Myogenic gene set shown in Figure 1F does not include many myogenic genes (except maybe Tnnt1), could the authors comment on why this is?

In the working model of fate decision, the bipotent cells have a signature of Myf5+, Pdgfra- and Pdgfa+. However, there is no strong evidence showing that these cells are Pdgfra-. Is there any cell that transiently express both Myf5 and Pdgfra mRNA or protein? What's the transcriptional expression level of Pdgfra in the scRNAseq data of cells sorted from mo

use Myf5GFP-P/+ embryos at E12.5 and E14.5? Is it expressed in the non-myogenic cluster? Additionally, for the EOM subset of the Myf5 lineage at E11.5, the expression plot of Myf5 is not shown. Does the cell population in between the myogenic and non-myogenic cells have both Myf5 and Pdgfra expression?

For immunofluorescence experiments, quantification of the proportion of double positive cells would be important. Especially the results verifying that Myf5 cells partially compensate for the lack of neural crest (Figure 3F-I'), since the green cells are hard to distinguish from the figure and the Myf5-expressing cells indicated by yellow arrowheads also exist in Wnt1 lineage cells.

Direct examination of Pdgfra+ cells using immunostaining or RNAscope in Myf5+ and Myf5-null embryos may be needed to strengthen the conclusion in Figure 5. In addition, the asterisk labeled β-gal+ cells in Figure 5D show neither myogenic marker nor *Sox9* expression. The fate of these cells may merit discussion.

What is shown in figure 6 is not very clear. More specifically, the fibrogenic fraction at stages 12.5 and 14.5 could be better described. What are the hatched boxes represented on the figure? It looks like the fibroblastic fraction strongly decreases with age which goes against their claim that the bipotential population is maintained.

A quantification of the Myod-iCre double positive cells shown on figure S5 would be helpful

Do the authors mean that the stages gave a similar level of enrichment for the same GO terms and reactome pathways?

The number of embryos stained by immunohistochemistry for each condition is on the low side (n=2) and could be increased.

Figure legends are too succinct and lack important details

In Figure 3, the labels of A-E are not consistent with the schematic depiction showing the positions of transverse sections.

*Reviewer #2 (Recommendations for the authors):*

A major strength of this manuscript is that it provides a comprehensive profiling of the embryonic craniofacial tissues in mouse, with the use of elaborated computational analysis, that will certainly be of great interest to the scientific community. The illustrations emanating from the in silico data are presented in an informative way, and the schemes are clear and elegant. It is also clearly written.

Specific point to address:

1. Some conclusions are solely based on the RNA-Seq data that would require further experimental validations. Specifically, based on Figure 4, the authors are claiming that two tyrosine kinase receptors, Bmpr1 and Ephb1 are among their top drivers genes of the myogenic EOM compartment, with two of their ligands expressed in the non-myogenic cells. To further test the relevance of an hypothetic paracrine signalling between these two Myf5 derived cell lineages, the authors could have undertaken a similar RNA in situ strategy (RNA-scope) as the one performed for Pdgfra and Pdgfa or IHC. A similar comment can be done with regards to figure 7, where the authors argue that Wnt/β-catenin pathway, emanating from the neural crest could influence the maintenance of the non-myogenic population. This hypothesis could be strengthen with downstream in situ validation experiments in the EOM region.

2. In figure 3, if I have understood correctly, it appears that the labels on the transversal sections depicted on the mouse embryo scheme are not reflecting those corresponding to the immunostaining tiles?

3. In figure 3 panel J, the authors are quantifying the number of double positive Tomato/GFP cells over the total GFP cells per region to determine the contribution of Myf5-lineage cells to the associated connective tissue in the EOM at E14.5. I have found it confusing that in order to create their graph, the authors are coming back to their transgenic line Myf5Cre/+;R26TdTom/+;PdgfraH2BGFP/+ without mentioning in the text, when they have previously used different lineage tracing strategy in the EOM at 13.5 (panel F to I).*Reviewer #3 (Recommendations for the authors):*

– Is it expected to obtain anterior somite cells in Mesp1+ lineage? Or even a lung/foregut population? Unexpected populations in Mesp1+ lineage are worth discussing already in the Results section, with respect to previous lineage tracing and clonal analyses, which also used the Mesp1 driver.

– P.4, L.111: "indicated" should read "suggested", because RNA velocity is consistent with the clonal interpretation but cannot be treated as conclusive. Proper clonal analysis would be necessary.

– Figure 2 is potentially interesting, but requires validation to be conclusive, both regarding the existence of bipotent progenitors, and the transition from a myogenic to a connective state.

– Statement that " Myf5-derived lineage contributes to connective tissue cells in the absence of neural crest" suggest that experimental removal of NCCs would trigger cartilage formation from myogenic lineages even in the ventral regions. Could this be tested?

– Figure 4A,B shows again excessive reliance on RNA velocity for clonal inference.

– P.6, L.178L BMPRI are serine/threonine kinases, not tyrosine kinase. The conclusion relying solely on scRNA-seq data regarding signaling needs to be validated by complementary experiments. Many plots on Figure 4F are seemingly redundant and not informative.

– Figure 4: if the idea is to reconstruct the EOM trajectories, why not integrate the data from distinct time points?

– Figure 5: SCENIC analysis is presented rather superficially. For instance, what genomic sequences were used to search for putative binding motifs? Is accessibility data for EOM cells available?

– Some of the non-myogenic factors mentioned (Ebf1, Six2, Foxc1) have also been involved in muscle development in other systems, so this point deserves to be clarified.

– The Myf5 null mutant showing an increase in *Sox9*+ cells is consistent with the hypothetized fate choice, but should be contrasted with the presumed signaling cross-talk between cell populations: if the muscles are missing, how does this affect their connective partners?

– Conclusions based on colocalization would be better supported by quantitative data e.g. % of double positive cells measured from images, instead of qualitative image accompanied by statement like "only rare double positive cells".

– It is not clear to this reviewer what the true value of Figures 6 and 7 are without further experimental validations. For instance, identification of candidate drivers for the non-myogenic fate calls for some degree of experimental validation of this prediction. the section on these TFs and on the Wnt pathway, inasmuch as they rely exclusively on computational analysis of descriptive scRNA-seq data, remain largely speculative.

---

## [Author Response]

Essential revisions:The reviewers found the paper interesting and potentially acceptable after revisions, including better descriptions of the scRNAseq by refining some of the in silico analyses and providing better quantitation. Currently, there is too much emphasis placed on possible signaling cross-talk which are not validated. While functional validation would be helpful, this seems like a minor aspect of the paper which could be reduced and toned down. The main contribution is the identification of myf5 positive progenitors able to contribute to the fibroblastic lineage. The full comments of the reviewers are provided below to help you in revising the paper.

We thank the Editor and Reviewers for assessing our work. We have added more explicit data on the scRNAseq, cell fate transition (Figure 4C) and markers used (Figure 1—figure supplement 3B, Figure 4—figure supplement 1D, Figure 6—figure supplement 1D and Figure 6—figure supplement 2C) to help better illustrate the myogenic and non-myogenic trajectories. We have also validated the signaling cross-talk directly by RNAscope (Figure 4E-H) and the expression of Foxp2 by immunostaining (Figure 7B-E). Assessment of the head, trunk and limb Myf5+ lineage potential to connective tissue was also performed by FACS (Figure 7F-G). In light of recently published work on the vascular potential of myogenic cells, we have added *Scube1* RNAscope analysis (Figure 5—figure supplement 1B-E). We have also consolidated our in-situ data with more replicates and quantifications (Figures 4H, 5J).

Reviewer #1 (Recommendations for the authors):Overall, this is a very interesting study and the conclusions of the paper are consistent with their presented data. The work will be of interest for the readership of eLife.There are some concerns I have that require some additional clarification:Overall, the description of the scRNAseq and the velocity analysis could be improved and for instance the criteria to decide whether a population is myo or fibrogenic should be better explained in the text.

The myogenic and non-myogenic fractions were defined using Pdgf dichotomy and expression of selected markers. Expression patterns of these relevant myogenic and fibrogenic genes that helped to define the myogenic and non-myogenic clusters have been added as supplemental data (Figure 1—figure supplement 3B, Figure 4—figure supplement 1D, Figure 6—figure supplement 1D and Figure 6—figure supplement 2C).

In figure 1B, lung and foregut precursors are identified whereas the population was sorted based on Mesp1 previous expression which labels the mesoderm. Are these mesodermal components of the lung and foregut?

We thank the reviewer for pointing out this feature that needed explanation. Indeed, this is the case and the text was edited to clarify this point.

The myogenic nature of the cells could be better described and illustrated on figure 1 and 2 and the expression pattern of myf5 in the different clusters would help. It would also be helpful to show the expression pattern of PDGFRA on the UMAPs as this is a key gene in this study. The Myogenic gene set shown in Figure 1F does not include many myogenic genes (except maybe Tnnt1), could the authors comment on why this is?

Expression patterns of relevant myogenic and fibrogenic genes (including Pdfra) were added as supplemental data (Figure 1—figure supplement 3B, Figure 4—figure supplement 1D, Figure 6—figure supplement 1D and Figure 6—figure supplement 2C) to strengthen the points made. In addition, we find *Msc, Des, Tcf21, Ttn, Myod* and *Myf5* as top 7, 27, 28, 29, 38 and 61 gene, respectively. A possible explanation for typical myogenic genes appearing further down the list is their salt and pepper expression pattern and resolution of the scRNAseq at this stage.

In the working model of fate decision, the bipotent cells have a signature of Myf5+, Pdgfra- and Pdgfa+. However, there is no strong evidence showing that these cells are Pdgfra-. Is there any cell that transiently express both Myf5 and Pdgfra mRNA or protein? What's the transcriptional expression level of Pdgfra in the scRNAseq data of cells sorted from mouse Myf5GFP-P/+ embryos at E12.5 and E14.5? Is it expressed in the non-myogenic cluster? Additionally, for the EOM subset of the Myf5 lineage at E11.5, the expression plot of Myf5 is not shown. Does the cell population in between the myogenic and non-myogenic cells have both Myf5 and Pdgfra expression?

To specifically address the question of *Myf5* and *Pdgfra* co-expression, we added the expression patterns of *Myf5* and *Pdgfra* in all datasets in Figure 1—figure supplement 3B, Figure 4—figure supplement 1D, Figure 6—figure supplement 1D and Figure 6—figure supplement 2C. Myf5+ cells are overwhelmingly Pdgfra- in all datasets. Myf5+/Pdgfra+ cells represent 8% (E10.5), 5,5% (E11.5), 0,5% (E12.5) and 0,15% (E14.5) of all cells in each dataset. In terms of Pdgfra output, Pdgfra+ cells represent 40% (E10.5), 56% (E11.5), 3% (E12.5) and 7% (E14.5) in each dataset. Cells at the transition between the myogenic and non-myogenic fractions express low-levels of myogenic genes (including Myf5) and low levels of non-myogenic genes (including *Pdgfra*) as illustrated by Figure 4C (heatmap). This information was added to the text (Line 182) and to the legend of Figure 1—figure supplement 3B, Figure 4—figure supplement 1D, Figure 6—figure supplement 1D and Figure 6—figure supplement 2C. For immunofluorescence experiments, quantification of the proportion of double positive cells would be important. Especially the results verifying that Myf5 cells partially compensate for the lack of neural crest (Figure 3F-I'), since the green cells are hard to distinguish from the figure and the Myf5-expressing cells indicated by yellow arrowheads also exist in Wnt1 lineage cells.

We have not included histograms of quantifications since we were not able to detect any Wnt1-derived nlacZ+ cell in these immunostainings. All cells with recent Myf5 history at 13.5 were found in the Mesp1 lineage (100% versus 0% in Wnt1-derived). Regarding the busy nature of these immunostaining, we added red arrowhead to highlight the Wnt1-derived connective tissue, which is Myf5-. This is consistent with Myf5^Cre/+^;R26^TdTom/+^;Pdgfra^H2BGFP/+^ counting at E14.5 in the EOM revealing higher Myf5-derived connective tissue in mesodermal domains, and Mesp1-driven scRNAseq which uncovered this population. However, future work could make use of Wnt1^Cre/+^;R26^TdTom/+^;Myf5^GFP/+^ and Mesp1^Cre/+^;R26^TdTom/+^;Myf5^GFP/+^ lines and FACS analyses at later stages to further exclude the possibility of Myf5+ cells deriving from neural crest, albeit neural crest generally accepted not to give rise to skeletal muscle.

Direct examination of Pdgfra+ cells using immunostaining or RNAscope in Myf5+ and Myf5-null embryos may be needed to strengthen the conclusion in Figure 5. In addition, the asterisk labeled β-gal+ cells in Figure 5D show neither myogenic marker nor Sox9 expression. The fate of these cells may merit discussion.

We have not investigated the fate of Myf5-derived cells in the absence of Myf5 beyond their integration in EOM cartilage primordium and *Sox9* expression. However, we reported previously that these cells can give rise to cartilage and dermis (Tajbakhsh 1996). Moreover, the identity of some Myf5-derived cells in the EOM in our study (those adjacent to the asterisk in Figure 5C) are indeed unknown. Given that they express neither *Sox9* nor myogenic markers, these cells might maintain an undifferentiated state at that stage (E12.5). It is also possible that Myf5 null cells are undergoing apoptosis as we reported previously (Sambasivan 2009). We added this point in the discussion.

What is shown in figure 6 is not very clear. More specifically, the fibrogenic fraction at stages 12.5 and 14.5 could be better described. What are the hatched boxes represented on the figure? It looks like the fibroblastic fraction strongly decreases with age which goes against their claim that the bipotential population is maintained.

The dotted boxes highlight the transitions to non-myogenic clusters in each dataset (Figure legend was updated). Indeed, the fibroblastic fraction is strongly reduced in the E12.5 and E14.5 datasets, which is expected, since they were generated using Myf5^GFP^, a recent history labeling strategy (compared to a cell lineage tracing with *Myf5^Cre^:R26^mTmG^*). The myogenic and non-myogenic cells were still noted in these datasets and RNA velocity suggested that this transition still occurs at these stages. However, it is also expected that this potential subsides as muscle development proceeds, and the need of myogenic progenitors for mesenchymal support reduces. The text was revised to clarify that point.

A quantification of the Myod-iCre double positive cells shown on figure S5 would be helpful

Quantification of Myod^iCre^ experiments has now been added and moved to Figure 5.

The sentence on line 229 and 230 is unclear. Do the authors mean that the stages gave a similar level of enrichment for the same GO terms and reactome pathways?

Yes, the number of genes underlying a given term is similar in all stages, as all stages contribute equally to a given term. The text was edited to clarify.

The number of embryos stained by immunohistochemistry for each condition is on the low side (n=2) and could be increased.

The n number was increased and quantifications were provided for RNA scope (n=3 for *Ephr*, *Bmp* and *Pdgf*, n=2 for *Scube1*), *Myod* lineage experiments (n=4), and *Foxp2* immunostainings (n=3).

Figure legends are too succinct and lack important details

Figure legends have been revised to include more information.

In Figure 3, the labels of A-E are not consistent with the schematic depiction showing the positions of transverse sections.

We thank the Reviewer for spotting this oversight which has now been corrected.

Reviewer #2 (Recommendations for the authors):A major strength of this manuscript is that it provides a comprehensive profiling of the embryonic craniofacial tissues in mouse, with the use of elaborated computational analysis, that will certainly be of great interest to the scientific community. The illustrations emanating from the in silico data are presented in an informative way, and the schemes are clear and elegant. It is also clearly written.Specific point to address:1. Some conclusions are solely based on the RNA-Seq data that would require further experimental validations. Specifically, based on Figure 4, the authors are claiming that two tyrosine kinase receptors, Bmpr1 and Ephb1 are among their top drivers genes of the myogenic EOM compartment, with two of their ligands expressed in the non-myogenic cells. To further test the relevance of an hypothetic paracrine signalling between these two Myf5 derived cell lineages, the authors could have undertaken a similar RNA in situ strategy (RNA-scope) as the one performed for Pdgfra and Pdgfa or IHC.

The signaling crosstalk inferred from the scRNAseq has now been validated by RNAscope and quantified (n=3 embryos). This data was added to Figure 4.

A similar comment can be done with regards to figure 7, where the authors argue that Wnt/β-catenin pathway, emanating from the neural crest could influence the maintenance of the non-myogenic population. This hypothesis could be strengthen with downstream in situ validation experiments in the EOM region.

This data has been moved to Figure 7—figure supplement 1E-H as it is more speculative as suggested by the Editor. Further comparative studies assessing NCC-derived and cranial mesoderm-derived signals could yield interesting results in regard to the orchestration of cranial muscle morphogenesis. This is however out of the scope of this study.

2. In figure 3, if I have understood correctly, it appears that the labels on the transversal sections depicted on the mouse embryo scheme are not reflecting those corresponding to the immunostaining tiles?

We thank the Reviewer for spotting this oversight which has now been corrected.

3. In figure 3 panel J, the authors are quantifying the number of double positive Tomato/GFP cells over the total GFP cells per region to determine the contribution of Myf5-lineage cells to the associated connective tissue in the EOM at E14.5. I have found it confusing that in order to create their graph, the authors are coming back to their transgenic line Myf5Cre/+;R26TdTom/+;PdgfraH2BGFP/+ without mentioning in the text, when they have previously used different lineage tracing strategy in the EOM at 13.5 (panel F to I).

We thank the Reviewer for spotting this oversight in displaying the data. This has now been clarified in the text to mention the mouse line being used.

Reviewer #3 (Recommendations for the authors):– Is it expected to obtain anterior somite cells in Mesp1+ lineage?

This point was demonstrated by Heude et al., and the text was edited to make it clearer to the readers.

Or even a lung/foregut population? Unexpected populations in Mesp1+ lineage are worth discussing already in the Results section, with respect to previous lineage tracing and clonal analyses, which also used the Mesp1 driver.

This issue was raised by another Reviewer and indeed it needed clarification in the text. The text was revised to include more information on Mesp1 derivatives.

– "indicated" should read "suggested", because RNA velocity is consistent with the clonal interpretation but cannot be treated as conclusive. Proper clonal analysis would be necessary.

The text was edited accordingly.

– Figure 2 is potentially interesting, but requires validation to be conclusive, both regarding the existence of bipotent progenitors, and the transition from a myogenic to a connective state.– Statement that " Myf5-derived lineage contributes to connective tissue cells in the absence of neural crest" suggest that experimental removal of NCCs would trigger cartilage formation from myogenic lineages even in the ventral regions. Could this be tested?

We agree with the Reviewer’s comment, however ablation of neural crest by diphtheria toxin induction for example (through a Wnt1 or Sox10 driver) would result in massive craniofacial defects, making any conclusion on the fate of myogenic cells challenging to interpret. While a more local disruption using regionalized Cre expression may be possible, we have not tried that approach as the combinatorial genetic strains are not available.

– Figure 4A,B shows again excessive reliance on RNA velocity for clonal inference.

We have added a coexpression score module to highlight the existence of a transition continuum between the 2 cell fates as was reported previously for neural crest fate transitions (Kameneva 2021).

– P.6, L.178L BMPRI are serine/threonine kinases, not tyrosine kinase. The conclusion relying solely on scRNA-seq data regarding signaling needs to be validated by complementary experiments. Many plots on Figure 4F are seemingly redundant and not informative.

We thank the Reviewer for spotting this oversight which has been corrected (and throughout the text). Unnecessary plots have been also removed.

– Figure 4: if the idea is to reconstruct the EOM trajectories, why not integrate the data from distinct time points?

We have tried to address this point, however merging the various EOM datasets appeared to be more challenging than previously anticipated. Therefore, we decided to keep them separated. New developments and approaches for batch effect corrections may yield better results in the future.

– Figure 5: SCENIC analysis is presented rather superficially. For instance, what genomic sequences were used to search for putative binding motifs? Is accessibility data for EOM cells available?

The genomic sequences and databases used have now been added to the Methods section. 2 cisTarget databases were used for SCENIC: “mm9-500bp-upstream-7species.mc9nr” (500bp upstream of TSS) and “mm9-tss-centered-10kb-7species.mc9nr” (10kb+/- TSS). The regulatory inferences are based on putative binding motifs and do not stem from EOM-specific interactions. However, they are curated based on coexpression in our EOM dataset. To our knowledge, accessibility data for the EOM specifically is not available.

– Some of the non-myogenic factors mentioned (Ebf1, Six2, Foxc1) have also been involved in muscle development in other systems, so this point deserves to be clarified.

We thank the Reviewer for this comment. A section has now been added to the discussion to discuss the Fox transcription factor family.

– The Myf5 null mutant showing an increase in Sox9+ cells is consistent with the hypothetized fate choice, but should be contrasted with the presumed signaling cross-talk between cell populations: if the muscles are missing, how does this affect their connective partners?

This is an interesting point we have not addressed. However, disruption of *Myf5* seems to have a strong effect on the commitment of a portion of the *Myf5*-derived cells to the cartilage lineage. The remaining cells (adjacent to the asterisk in Figure 5C), which do not express *Sox9* or myogenic factors, may be undergoing apoptosis as we described previously for the *Myf5* null (Sambasivan 2009).

– Conclusions based on colocalization would be better supported by quantitative data e.g. % of double positive cells measured from images, instead of qualitative image accompanied by statement like "only rare double positive cells".

Quantification of the Myod^iCre^ experiments have now been added to Figure 5 (n=4).

– It is not clear to this reviewer what the true value of Figures 6 and 7 are without further experimental validations. For instance, identification of candidate drivers for the non-myogenic fate calls for some degree of experimental validation of this prediction. the section on these TFs and on the Wnt pathway, inasmuch as they rely exclusively on computational analysis of descriptive scRNA-seq data, remain largely speculative.

Foxp2 expression has now been validated by immunostaining and quantified (Figure 7). The involvement of Wnt/Bcat activity was move to Figure 7—figure supplement 1E-H as it is more speculativ